# Understanding the Limits of Deep Tabular Methods with Temporal Shift

**Hao-Run Cai** [1 2]   **Han-Jia Ye** [1 2]

## Abstract

Deep tabular models have demonstrated remarkable success on *i.i.d.* data, excelling in a variety of structured data tasks. However, their performance often deteriorates under temporal distribution shifts, where trends and periodic patterns are present in the evolving data distribution over time. In this paper, we explore the underlying reasons for this failure in capturing temporal dependencies. We begin by investigating the training protocol, revealing a key issue in how model selection performs. While existing approaches use temporal ordering for splitting validation set, we show that even a random split can significantly improve model performance. By minimizing the time lag between training data and test time, while reducing the bias in validation, our proposed training protocol significantly improves generalization across various methods. Furthermore, we analyze how temporal data affects deep tabular representations, uncovering that these models often fail to capture crucial periodic and trend information. To address this gap, we introduce a plug-and-play temporal embedding method based on Fourier series expansion to learn and incorporate temporal patterns, offering an adaptive approach to handle temporal shifts. Our experiments demonstrate that this temporal embedding, combined with the improved training protocol, provides a more effective and robust framework for learning from temporal tabular data.

## 1. Introduction

Tabular data is one of the most prevalent data formats in a wide range of real-world applications, such as e-commerce (Nederstigt et al., 2014) and healthcare (Hassan et al., 2020).

[1]School of Artificial Intelligence, Nanjing University, China [2]National Key Laboratory for Novel Software Technology, Nanjing University, China. Correspondence to: Han-Jia Ye <yehj@lamda.nju.edu.cn>.

*Proceedings of the 42$^{nd}$ International Conference on Machine Learning*, Vancouver, Canada. PMLR 267, 2025. Copyright 2025 by the author(s).

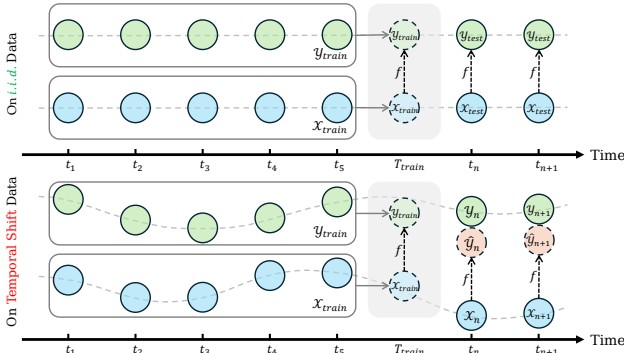

*Figure 1.* **Illustration of the challenges posed by temporal shifts.** The change in data distribution over time is represented by dots on a line, with the dashed line depicting the underlying data distribution at different time slices. The shaded box indicates the mapping $f$ learned from the training data at $T_{\text{train}}$, while the training data is typically treated as *i.i.d.* on $\mathcal{X}_{\text{train}}$ and $\mathcal{Y}_{\text{train}}$ in classical training processes. On *i.i.d.* data, the model can directly apply the learned mapping $f$ to make accurate predictions on test data, but it fails to generalize effectively when temporal shifts occur.

It consists of instances (rows) and features (columns), where the label can either be categorical or continuous, corresponding to classification and regression tasks, respectively. The goal is to learn a mapping strategy from features to labels that can be directly applied to independent and identically distributed (*i.i.d.*) test data for accurate predictions (Bishop, 2007; Mohri et al., 2012). While tree-based methods remain powerful (Breiman, 2001; Chen & Guestrin, 2016; Ke et al., 2017; Prokhorenkova et al., 2018), recent advancements in deep learning methods have demonstrated promising results (Klambauer et al., 2017; Arik & Pfister, 2021; Wang et al., 2021b; Gorishniy et al., 2021; 2024; 2025; Hollmann et al., 2023; 2025; Ye et al., 2023; 2025; Holzmüller et al., 2024; Jiang et al., 2024; 2025; Liu & Ye, 2025; Liu et al., 2025).

Most machine learning approaches are built on the assumption of *i.i.d.* data. However, in open environments (Zhou, 2022), distribution shifts (Gardner et al., 2023; Tschalzev et al., 2024) frequently occur between the training and testing data, which can manifest as shifts in the feature space, label space, or the mapping between them. Moreover, in practical applications, data often exhibits temporal distribution shifts (Rubachev et al., 2025), a particularly common type of distribution shift that introduces additional challenges: instead of just shifts between training and test sets, temporal shifts can occur within the training set or the test

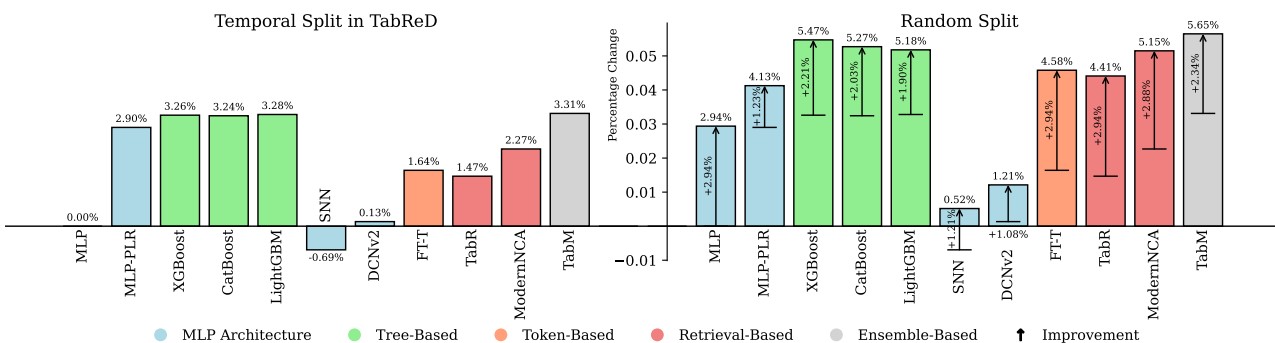

*Figure 2.* **Performance comparison between temporal split in Rubachev et al. (2025) and random split on TabReD benchmark**, where only the data splitting strategy before $T_{\text{train}}$ is changed. The percentage change represents the *robust average* of performance difference compared to the *MLP with original temporal split*. A positive percentage change indicates that the method outperforms the MLP with temporal split. **Left:** We reproduce the experiment from Rubachev et al. (2025) and ensure a fair comparison by removing additional numerical feature encodings, as explained in Appendix A. In this setting, the performance of retrieval-based methods significantly declines, falling behind tree-based methods and MLP-PLR, while TabM achieves the best performance. **Right:** Performance improvements observed under the random split. Retrieval-based methods show the most substantial gains, and model rankings align more closely with conventional expectations. Detailed results are provided in Appendix C.1.

set itself. For example, in house price prediction (Matveev & Sidorova, 2017), the task is to predict future house prices using historical transaction data, which includes features such as location, neighborhood conditions, and economic factors. However, in actual practice, the trend of housing policies or the periodic fluctuations of public sentiment can also play crucial roles in influencing house prices.

These temporal shifts lead to the failure of the model when the training process assumes the data to be *i.i.d*. The challenge of training on temporal shift data is illustrated in Figure 1. Formally, we aim to perform model training and validation using data prior to $T_{\text{train}}$, and subsequently use the trained model to make predictions on data after $T_{\text{train}}$. Once the temporal shift occurs, it leads to discrepancies between the mapping learned during training and the one required for deployment, making the model ineffective.

Since temporal shifts are common and present significant challenges, we have turned our attention on whether tabular data models that perform well in classic *i.i.d*. scenarios can effectively manage temporal shifts. In recent studies, Rubachev et al. (2025) introduced the TabReD benchmark. By comparing the performance differences of various methods on the original temporal shift dataset and the *i.i.d*. dataset constructed through shuffling, their observations revealed that while models with MLP architectures exhibit relative robustness during temporal shifts, retrieval-based methods, which are highly competitive in current benchmarks (Ye et al., 2024), suffer a marked performance degradation in temporal shift scenarios. This observation demonstrates that the occurrence of temporal shifts can introduce significant biases in the evaluation of model performance, further emphasizing the importance of understanding and managing these temporal shifts.

In temporal shift tasks, the absence of accurate test data

validation during the training stage (Blanchard et al., 2021) makes model selection more impactful, as deep learning models are optimized epoch-wise. TabReD employs a temporal splitting strategy to match the test scenario, utilizing earlier data for training and later data prior to $T_{\text{train}}$ as the validation set for model selection. Surprisingly, we discovered that by merely altering the splitting strategy on the same training data, even when randomly splitting the training and validation sets and ignoring temporal order, the model outperformed the temporal split, resulting in significantly enhanced performance, as illustrated in Figure 2. Moreover, the performance rankings of the models became more consistent with established results on *i.i.d*. data, with retrieval-based methods showing more distinct improvements. This unexpected finding underscores the urgent need to investigate effective data splitting strategies to accurately assess model performance and reveal its true capabilities in the presence of temporal shifts. Despite the performance improvement, the random split exhibits pronounced instability in temporal shift context, which also requires attention.

We begin by **analyzing the factors contributing to the ineffectiveness of temporal splitting in this context**. While temporal-based splitting is commonly employed in forecasting tasks to maintain causal relationships (Bergmeir & Benítez, 2012), the most intuitive difference when adopting a temporal split in tabular learning lies in the *time lag between the training and test sets*, and the *bias in validation*, since the data closest to the test time are not used for training, and the shift degree of test set relative to validation set is more significant in temporal splits. By investigating the impact of these two factors, we find that minimizing the training lag concentrates the model's performance closer to the test time, while reducing validation bias by aligning the shift degree makes the model better generalize on test data. Building on these insights, we propose a training pro-

tocol along with an improved temporal split that leverages these advantages, resulting in a comparable performance to random splitting while providing better stability.

We further **investigate the impact of temporal shifts on deep tabular methods from the perspective of *feature representations***. Through the visualization of the model's deep-layer representations, we observe that the trends and periodic information, which are prevalent in the raw data, gradually diminish in the feature representations. This indicates a loss of temporal information in the representation, leading to the restriction of the model's temporal prediction ability. This also explains why existing methods encounter challenges in addressing temporal shifts.

Based on the analysis presented above, it is essential to effectively incorporate temporal information into the model. To address this gap, we **develop a lightweight, plug-and-play *temporal embedding* method** based on Fourier series expansion. This approach equips the model with learnable periodic and trend information while preserving its original capabilities. Experiments demonstrate that temporal embedding further enhances the capacity of all models to address shift problems. Furthermore, this method can be regarded as a *temporally adaptive approach*. Once the model is provided with temporal information, it can acquire knowledge specific to different temporal stages. This capability allows the model to adjust adaptively after deployment.

In summary, this paper investigates the challenges posed by temporal distribution shifts in tabular data and explores effective strategies to address them. Starting with a training protocol that fully leverages temporal data, we delve into the impact of temporal shifts during the model training process. We further propose a temporal embedding method to compensate for the loss of temporal information, thereby enhancing the model's ability to adapt to temporal shifts and improve its real-world performance. These insights offer crucial guidance for the future development of deep tabular methods in temporal shift scenario.

## 2. Related Work

### 2.1. Tabular Machine Learning

Tabular data is a common format across various applications, and the main solutions can be categorized into tree-based methods, token-based methods, retrieval-based methods, ensemble-based methods, and MLP architecture methods. Classical *tree-based* methods like Random Forest (Breiman, 2001), XGBoost (Chen & Guestrin, 2016), LightGBM (Ke et al., 2017), and CatBoost (Prokhorenkova et al., 2018), remain powerful and widely adopted in real-world scenarios. FT-Transformer (FT-T) (Gorishniy et al., 2021) is a *token-based* method that utilizes the transformer architecture, TabR (Gorishniy et al., 2024) and ModernNCA (Ye

et al., 2025) are *retrieval-based* methods that make predictions by retrieving neighbors in the representation space, TabM (Gorishniy et al., 2025) provides an *ensemble-based* method on MLP, while other methods like SNN (Klambauer et al., 2017), DCNv2 (Wang et al., 2021b) and MLP-PLR (Gorishniy et al., 2022) focus on improving the *MLP architecture*. With the continuous improvement of deep tabular models on established benchmarks (McElfresh et al., 2023; Ye et al., 2024), the practical deployment of such models has become a pressing consideration.

### 2.2. Distribution Shift in Tabular Data

Current research on distribution shifts can be broadly categorized into two main approaches. The first category focuses on scenarios in which target domain data is partially available. In these cases, transfer learning techniques are commonly employed to dynamically adjust model parameters during deployment by leveraging test-time data. TableShift (Gardner et al., 2023) applies various classical methods of domain generalization (Ganin et al., 2016) and domain adaptation (Sun & Saenko, 2016) into deep tabular learning, alongside the recently proposed test-time adaptation techniques for tabular data (Kim et al., 2024). While effective in certain contexts, these approaches often assume the availability of target domain information at test time, which may be infeasible in real-world settings. In our experimental setup, the test time information is entirely invisible during both the training and testing stage, thus methods in this category are not applicable to our setting.

The second category addresses situations in which target domain data is entirely unavailable, representing a more common and challenging scenario. Approaches within this category can be further divided into two types: those aimed at enhancing model robustness and those focused on the active learning of shift patterns. The first type seeks to improve the inherent robustness and generalization of models, thereby indirectly mitigating the impact of distribution shifts. For instance, Gorishniy et al. (2025) demonstrates the effectiveness of ensemble strategies in addressing distribution shifts in tabular data. Our exploration of the training protocol serves as an effective approach to enhancing model generalization performance. The second type incorporates knowledge of distribution shifts directly into the model through adaptive methods. One such approach is presented by Helli et al. (2024), which employs second-order models to capture and adapt to shifts based on learned causal relationships. However, methods in this category are heavily dependent on specific model architectures, such as PFN (Hollmann et al., 2023), and tend to be computationally expensive. Consequently, we did not include a comparison with this category of methods. In contrast, our temporal embedding method offers a lightweight and plug-and-play solution for achieving temporal adaptability.

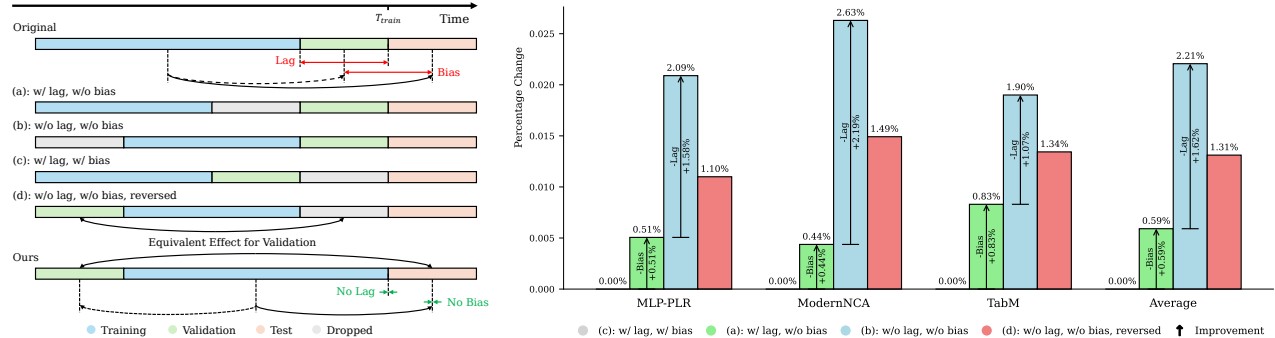

*Figure 3.* **Left:** *Experimental design* for temporal split strategies. The top panel shows the original baseline adopted by Rubachev et al. (2025). The middle panel (a)–(d) illustrates: (i) training lag (a *vs.* b; Section 4.1), (ii) validation bias (a *vs.* c; Section 4.2), and (iii) validation equivalence (b *vs.* d; Section 4.3). The bottom panel presents our proposed strategy (Section 4.4). **Right:** *Performance improvement* of different splitting strategies relative to split (c) on the TabReD benchmark, demonstrating benefits in reducing training lag and validation bias. Notably, the performance degradation from (b) to (d) is much smaller than the improvement achieved by (b), suggesting that adopting the alternative splitting strategy to maximizing data utilization is preferable. Detailed results in Appendix C.1.

While TableShift (Gardner et al., 2023) emphasizes domain-to-domain shifts, TabReD (Rubachev et al., 2025) introduces the concept of temporal shifts. They argue that all tabular data is inherently temporal, advocating for the use of temporal splits, especially in industrial applications where data is typically feature-rich and includes visible timestamps. In this study, we further investigate a training protocol and propose a more effective and robust framework for learning from temporal tabular data.

### 2.3. Distribution Shift in Other Domains

The study of distribution shifts originated in computer vision, with early research primarily focusing on transfer learning techniques to address domain-to-domain shifts, including domain generalization (Blanchard et al., 2011; 2021), domain adaptation (Ganin & Lempitsky, 2015), and test-time adaptation (Wang et al., 2021a). Later, the focus expanded to encompass continual distribution shifts, with strategies for adapting models to sequential domain changes (Wang et al., 2022), as well as addressing recurring (Hoang et al., 2024) and non-*i.i.d.* (Gong et al., 2022) temporal shifts. Wild-Time (Yao et al., 2022) explores real-world temporal shifts but translates them into domain-to-domain settings, neglecting temporal continuity. In comparison, these methods are primarily designed for images and most utilize transfer learning. Many image-based methods, when directly applied to tabular data, are considered to perform poorly (Gardner et al., 2023). This may be due to the more complex and diverse distribution shifts in tabular data, which involve greater temporal dependencies and continuity.

## 3. Preliminary

Generally, a tabular dataset with $n$ instance is represented as $\mathcal{D} = \{(\boldsymbol{x}_i, y_i)\}_{i=1}^n$, where $\boldsymbol{x}_i \in \mathcal{X}$ is the input feature and $y_i \in \mathcal{Y}$ is the target label. Here, $\mathcal{X}$ denotes the feature space

(*e.g.*, $\mathbb{R}^d$) and $\mathcal{Y}$ denotes the label space (*e.g.*, classes or real values). The goal is to learn a mapping $f : \mathcal{X} \to \mathcal{Y}$ that can generalize from the observed data to unseen instances, effectively predicting $\hat{y}_i = f(\boldsymbol{x}_i)$ for a given input $\boldsymbol{x}_i$.

The objective function $f$ can be decomposed into two components: $f = g \circ h$, where $g$ for feature extraction and $h$ for prediction. In *MLP architectures*, both $g$ and $h$ are neural networks, with $g$ consisting of multiple layers and $h$ being the final output layer. In *ensemble-based methods*, multiple models $f_i = g_i \circ h_i$ are trained, and the final prediction is obtained by averaging the outputs, which reduces variance and improves generalization. *Retrieval-based methods* use a neural network for $g$ and a non-parametric prediction method, such as a soft $k$-NN adopted in ModernNCA (Ye et al., 2025), for $h$, where predictions are based on the closest neighbors of "candidates" — instances from the training set that are mapped to the representation space and serve as potential reference points during prediction.

A temporal tabular dataset collected before $T_{\text{train}}$ for model training and deployment can be represented as $\mathcal{D}_{\text{trainval}} = \bigcup_{t \leq T_{\text{train}}} \mathcal{D}_t$, where $\mathcal{D}_t = \{(\boldsymbol{x}_i, y_i, t)\}_{i=1}^{n_t}$ denotes the set of $n_t$ instances collected at time $t$, each attached with its timestamp. $T_{\text{train}}$ is the time at which training is performed. After training, the model is deployed to an open environment and evaluated on $\mathcal{D}_{\text{test}} = \bigcup_{t > T_{\text{train}}} \mathcal{D}_t$. The training data $\mathcal{D}_{\text{trainval}}$ need to be further split into training and validation set in training stage. TabReD (Rubachev et al., 2025) adopts a temporal split where the data is divided at $T_{\text{val}}$, such that $\mathcal{D}_{\text{train}} = \bigcup_{t \leq T_{\text{val}}} \mathcal{D}_t$ and $\mathcal{D}_{\text{val}} = \bigcup_{T_{\text{val}} < t \leq T_{\text{train}}} \mathcal{D}_t$.

A *temporal distribution shift* refers to a specific type of distribution shift where the data is collected sequentially over time, and the underlying data distribution evolves as time progresses. Formally, let $\mathcal{X}_t$ and $\mathcal{Y}_t$ denote the feature space and label space at time $t$, respectively. For $\boldsymbol{x}_t \in \mathcal{X}_t$ and $y_t \in \mathcal{Y}_t$, a temporal distribution shift may involve

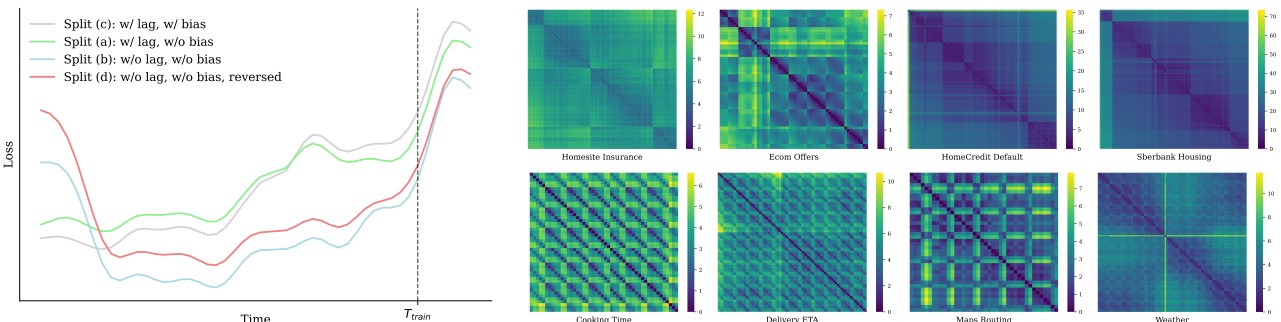

*Figure 4.* **Left:** The *loss distribution* shows how the model's performance distributed across time slices under different validation splitting strategies. The vertical axis represents the loss, where lower is better. Non-lagged splits (b) and (d) achieve better performance around $T_{\text{train}}$ compared to lagged splits (a) and (c), while the higher-biased split (c) performs better on training-available data but fails to generalize compared to the lower-biased split (a). The loss distribution is smoothed by a Gaussian filter for better visualization. **Right:** The *MMD heatmap* visualizing the distribution distance between different time slices using linear kernel. Time slices are divided by date.

changes in $p(\boldsymbol{x}_t) \neq p(\boldsymbol{x}_{t'})$, $p(y_t) \neq p(y_{t'})$, $p(y_t \mid \boldsymbol{x}_t) \neq p(y_{t'} \mid \boldsymbol{x}_{t'})$, or even $\mathcal{X}_t \neq \mathcal{X}_{t'}$ or $\mathcal{Y}_t \neq \mathcal{Y}_{t'}$, for some $t \neq t'$.

Conventional approaches for tabular data analysis have primarily relied on the *i.i.d.* assumption, where the feature extraction function $g$ typically maps all input data to a shared representation space, implicitly assuming that the data distribution remains stationary over time. As a result, the impact of temporal dynamics on the effectiveness of these methods remains largely unexplored and warrants investigation.

## 4. Why Temporal Split Ineffective?

The fundamental challenge in addressing temporal shifts stems from the inherent difficulty in characterizing evolving data distributions, which requires models to maintain robustness against unknown future variations. Temporal splits are frequently used in forecasting-related tasks to uphold the sequential dependencies inherent in the data (Bergmeir & Benítez, 2012; Zeng et al., 2023; Han et al., 2024), yet they generally show inferior performance compared to random splits in tabular data contexts, as evidenced in Section 1 and Figure 2. This counterintuitive phenomenon warrants systematic analysis through multiple perspectives: We first start with the most intuitive distinction between random split and temporal split, and verify the hypothesis of how training lag (Section 4.1) and validation bias (Section 4.2) impact the model performance. Then we discover the equivalence of the validation set in different temporal directions (Section 4.3). Building on this, we propose an enhanced splitting strategy (Section 4.4) for temporally shifted tabular data, which achieves performance comparable to random splitting while offering improved stability.

### 4.1. Training Lag

From the perspective of the training set, temporal split adopted in Rubachev et al. (2025) introduces a *temporal lag* between training and test set, while random split provides more instances closer to the test time for training. An intuitive hypothesis is that instances closer to the test time are more reliable, as the distribution near the test time tends to be more similar to the actual test-time distribution, and using them for training could have a greater effect than for validation. Hence we adapt a pair of splitting strategies, shown in Figure 3 left (a) and (b), where identical validation and test sets are maintained while varying the lag between training and test set, thereby isolating and quantifying the impact of training lag on model performance.

We selected MLP-PLR, ModernNCA, and TabM as representative methods for MLP architecture, retrieval-based, and ensemble-based methods, respectively, and conducted experiments of splitting strategy on these three method. Experimental results shown in Figure 3 right infers that this hypothesis holds for all three methods, with a total average improvement of 1.62%. Among them, the retrieval-based methods ModernNCA shows the highest improvement of 2.19%, indicating the importance of no-lag candidates for retrieval-based methods in the presence of temporal shifts.

### 4.2. Validation Bias

Current deep tabular methods rely on information from the validation set for model selection, as deep learning models are optimized epoch-wise during training. In the context of temporal shifts, this reliance becomes particularly problematic, since there is a lack of accurate validation of the test data in the training stage (Ganin & Lempitsky, 2015; Blanchard et al., 2021). The time gap between the training and test sets is larger than the gap between the training and validation sets in the previous temporal split, leading to a more significant distribution shift at test time. This makes it more challenging to accurately predict test-time instances compared to validation, thereby introducing *bias into the validation process*. Therefore we further design the splitting strategy shown in Figure 3 left (c), which shares the training and test sets with (a), but the latter has a more considerable

| Splits | MLP | PLR | FT-T | SNN | DCNv2 | TabR | MNCA | TabM | XGBoost | CatBoost | LGBM | Avg. Imp. |
|---|---|---|---|---|---|---|---|---|---|---|---|---|
| **Mean Performance ↑** | | | | | | | | | | | | |
| Random | +4.30% | +0.73% | +3.76% | +1.38% | +2.11% | +2.00% | +2.53% | +1.51% | +1.79% | +2.09% | +1.73% | +2.17% |
| Ours | +3.50% | +0.75% | +2.78% | +1.38% | +3.01% | +2.20% | +2.49% | +1.25% | +2.06% | +2.37% | +2.14% | **+2.18%** |
| **Standard Deviation ↓** | | | | | | | | | | | | |
| Random | +1.81% | +126% | +0.15% | +44.0% | +0.06% | +224% | +74.9% | +44.3% | +456% | +105% | +616% | +154% |
| Ours | +29.7% | +50.9% | +5.59% | +16.8% | +8.86% | +82.2% | −10.8% | +37.7% | −15.2% | −30.6% | +8.59% | **+16.7%** |
| **Performance Rankings ↓** | | | | | | | | | | | | |
| Random | 8.250 | 5.625 | 5.625 | 10.250 | 9.625 | 8.000 | 4.750 | **2.750** | 3.125 | 3.125 | 4.875 | – |
| Ours | 8.000 | 5.750 | 7.500 | 9.500 | 8.375 | 8.125 | 4.875 | 4.000 | 3.375 | **2.125** | 4.375 | – |

*Table 1.* **Comparison of performance and stability** between the random split in Figure 2 and our proposed temporal split in Section 4.4, measured by the *average percentage change* on the TabReD benchmark, along with the performance ranking of each method. "PLR," "MNCA," and "LGBM" denote "MLP-PLR," "ModernNCA," and "LightGBM," respectively. The percentage change represents the difference in the mean (higher is better) or the standard deviation (lower is better, indicating stability) of performance, relative to the original temporal split in Rubachev et al. (2025), for each method. The results show that our temporal splitting strategy achieves performance comparable to the random split, while offering significantly better stability. The ranking change is minimal, with token-based methods favoring the random split and tree-based methods performing better with our temporal split. The comparison of methods under random and temporal splits is also plotted in Figure 2 and Figure 7. Detailed results are provided in Appendix C.1, with extended rankings.

validation bias since the time interval difference between train-val and train-test is much larger.

The performance comparison of split (a) and (c), as shown in Figure 3 right, confirms that validation bias also has a notable impact on performance, especially for ensemble-based methods. The total average improvement is 0.59%, while the ensemble-based TabM show a more significant improvement of 0.83%. This is explainable since the ensemble-based methods are robust to the training data quality by reducing the variance, but sensitive to the bias of validation.

### 4.3. Bridging the Past and Future in Temporal Split

Through the above exploration, we have confirmed the improvements in model performance achieved by reducing the training lag and validation bias. The main question now becomes how to utilize the above insights.

To illustrate how reducing the training lag and validation bias improves model performance, we visualize the loss distribution of the model across different time slices under different validation splitting strategies. Figure 4 left shows the loss distribution for the CT dataset using MLP-PLR model, it clearly shows how reducing training lag and validation bias improves the model performance. Each splits achieve the best performance among the training-available time slices, which allows the model to perform better on instances closer to the test time in the non-lagged split (b) compared to the lagged splits (a) and (c). Furthermore, the higher-biased split (c) works better on training splits but struggles on the test splits, while the lower-biased split (a) achieves a more balanced performance across training and test splits. This indicates that a lower-biased validation set brings a more precise direction of the test-time distribution, enabling the model to generalize better. The above observation again enhances the insights of training lag and

validation bias. On a deeper level, reducing the training lag concentrates the model's performance around a time point closer to the test time, while precise validation ensures that the model's performance can effectively generalize from this concentrated time point to the test period.

In addition, another interesting observation in Figure 4 left is that the model in split (b) never meet the most former data splits during training and validation, but achieves a relatively well performance on it compared to the test splits. Building on the insight that the validation set is mainly for maintaining generalization performance, this finding inspired us to explore whether data from the opposite temporal direction could be an effective validation set, which is rely on the assumption that the temporal shifts distributed uniformly across each time slices. We further use a heatmap to visualize the distribution distance between different time slices using Maximum Mean Discrepancy (MMD) (Gretton et al., 2006), as shown in Figure 4 right. The MMD heatmap reveals that all datasets exhibit temporal shifts characterized by trends and multiple periodic components. The regular diagonal stripes both indicate the presence of periodicity, and suggest that the sample distributions at identical time intervals are similar, which confirms the empirical uniformity of temporal shifts across time slices.

Finally we design the splitting strategy shown in Figure 3 left (d), where the validation set is in the opposite temporal direction compared to split (b), meanwhile maintaining the training lag and validation bias. The results in Figure 3 right indicate a performance drop of 0.91% compared to split (b). However, it is important to highlight that, this performance degradation is noticeably smaller than the improvement observed in (b) relative to (a). This indicates that adopting this alternative splitting strategy to minimize the training gap is always a desirable approach.

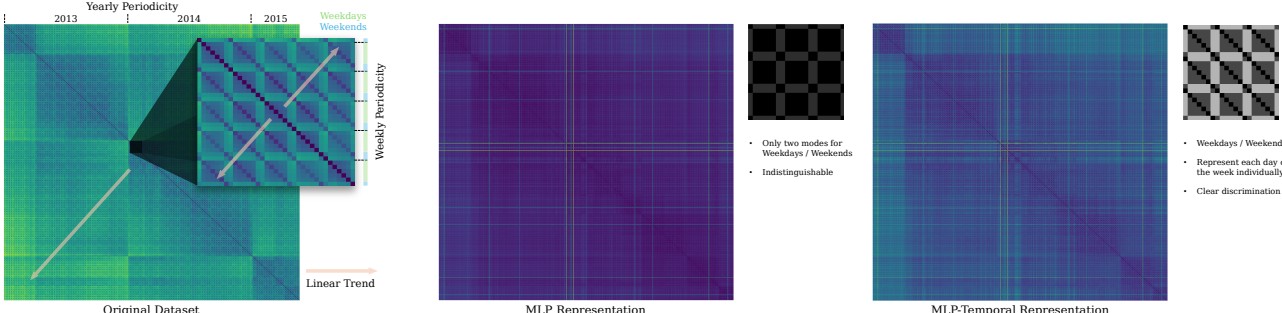

*Figure 5.* **Left:** Detailed MMD heatmap of the HI dataset, illustrating both *trend* (lighter colors farther from the diagonal) and yearly/weekly *periodicity* (stripes at different scales) in the data. **Middle** and **Right:** MMD heatmaps of the representations learned by an MLP *before and after applying our temporal embedding*. Without temporal embedding, the model captures only coarse-grained patterns (*e.g.*, weekday *vs.* weekend) with weak discrimination. After incorporating temporal embedding, the learned representations align with the data distribution, capturing phase-specific temporal feature (*e.g.*, day of the week) and achieving clear distinction. See Appendix C.4 for more results.

In detail, the model performance on this splitting strategy is highly dependent on the dataset, which is primarily due to the nonuniform temporal sampling in some datasets, where maintaining the same amount of validation data compromises the consistency of the time intervals. Further explanation can be found in Appendix A.

### 4.4. Our Temporal Split

Based on the above findings, we introduce the following training protocol for temporal tabular data:

1. The lag between training and test set should be minimized since instances near the test time are more valuable for training, rather than for model selection.
2. The validation bias should be minimized, which can be achieved by reducing the time interval difference between train-val and train-test.
3. The equivalence property of the validation set in different temporal direction is maintained for most tasks. An effective validation is available in the opposite temporal direction by aligning the degree of shift in the validation set with the actual shift between training and testing data.

We further propose a more effective temporal splitting strategy that fully leverages this protocol, shown in the bottom of Figure 3 left. In this strategy, the training lag is minimized to zero, and the validation set is also aligned with the test set, as it has a similar time interval relative to the training set thus exhibits a similar degree of distribution shift.

The performance and stability comparison between the random split and our newly proposed temporal split is presented in Table 1. The results demonstrate that our splitting strategy achieves a performance improvement comparable to the random split (2.18% vs. 2.17% on average) relative to the baseline temporal split in Rubachev et al. (2025). Additionally, while our temporal split shows a modest increase in the standard deviation of performance scores (AUC for classification and RMSE for regression) by 16.69%, the random

split results in a much larger increase of 153.81%. This indicates that, while both methods achieve similar performance gains, our temporal split significantly outperforms the random split in terms of stability. A comparison of stability using the robustness score is provided in Appendix C.1.

## 5. What is Lost in Temporal Training?

Looking back to the MMD heatmap in Figure 4 right, we observe that the original data distribution offers a rich source of temporal information, including the periodicity and the trend. Figure 5 left presents a detailed MMD heatmap visualization of the HI dataset, revealing both yearly and weekly periodicity, as well as the underlying trend. We further investigate the impact of temporal shifts on deep tabular methods from the perspective of feature representations.

Unexpectedly, by comparing the MMD heatmaps of the learned representations of MLP in our training protocol (shown in Figure 5 mid), we observe that the periodicity and the trend are lost in the model representations. Instead of the diagonal stripes observed in the original data distribution, the learned representations exhibit only shallow grids and a more uniform distribution, suggesting that temporal information has not been effectively preserved. This indicates that the model has captured the distinction between weekdays and weekends but failed to capture the long-term periodicity and finer details of short-term cycles.

This phenomenon may account for the suboptimal performance of datasets with clear periodic patterns, such as CT, DE, and MR datasets, which are socially related and exhibit distinct weekly cycles, while the WE dataset, being influenced by natural cycles, shows a clear yearly pattern. We would expect models using temporal splits to capture long-term periodicity and trends, enabling them to learn extrapolative knowledge. However, this critical knowledge does not seem to be effectively learned. In contrast, random splits appear more proficient at capturing local patterns, par-

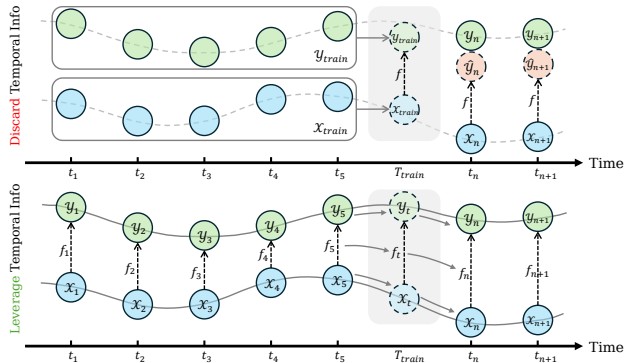

*Figure 6.* **Illustration of adaptive approaches for temporal shift mitigation.** Our proposed temporal embedding method offers a lightweight alternative that implicitly enables adaptation by embedding temporal information. This allows the model to learn phase-specific knowledge and adjust the mapping $f_t$ accordingly, thereby achieving temporal generalization.

| Emb. | MLP | MLP-PLR | ModernNCA | Avg. Imp. |
|------|-----|---------|-----------|-----------|
| Num  | $-0.04\%$ | $-0.06\%$ | $-0.04\%$ | $-0.05\%$ |
| Time | $-0.70\%$ | $-0.15\%$ | $-0.32\%$ | $-0.39\%$ |
| PLR  | $+0.70\%$ | $+\mathbf{0.01\%}$ | $+0.02\%$ | $+0.25\%$ |
| Ours | $+\mathbf{1.31\%}$ | $+\mathbf{0.01\%}$ | $+\mathbf{0.30\%}$ | $+\mathbf{0.54\%}$ |

*Table 2.* **Performance gain with temporal embeddings** on the TabReD benchmark. Embeddings are applied only to timestamps, which are then treated as numerical features for the model. Non-learnable embeddings yield limited improvement; the learnable PLR variant helps MLP slightly. Our temporal embedding consistently outperforms all others. See Appendix C.2 for details.

ticularly those associated with short-term periodicity. This could explain why temporal splitting does not consistently outperform random splitting in these cases. Furthermore, it also explains the reasons why existing methods encounter challenges in dealing with temporal shifts.

## 6. Temporal Embeddings

Building on the above analysis, it is essential to incorporate temporal information into the model in a manner that effectively captures the underlying temporal dependencies. To address this issue, we propose a *lightweight, plug-and-play temporal embedding method* specifically designed for timestamps, aiming to investigate whether providing explicit temporal information through embedding can lead to performance improvements. Empirically, timestamps are often treated as noise and discarded. However, in the context of temporal shifts and temporal splitting, timestamps likely contain crucial temporal information that enables the model to align with the periodicity and trends inherent in the temporal sequence. Existing works have also emphasized that, in certain contexts, timestamps serve as a valuable feature (Wang et al., 2024; Zeng et al., 2024).

Our temporal embedding also serves as an *adaptive model-*

*based approach*, as the model can learn temporal stage-specific knowledge by leveraging timestamp information, thereby adaptively adjusting its mapping at different temporal stages after deployment. Formally, the model's objective function can be written as $f = g \circ h$, where $g$ maps the input features to the same representation space. When the data exhibits temporal shifts, this implies that there exists a specific mapping $f_t = g_t \circ h_t$ at time step $t$. The model can now learn phase-specific knowledge and adjust the mapping $f_t$ accordingly, as illustrated in Figure 6.

In our embedding method, we fit multi-scale periodicity using Fourier series expansion (Tancik et al., 2020; Li et al., 2021). The Fourier series can be expressed as:

$$f(t) = \sum_{k=1}^{\infty} a_k \sin\left(\frac{2\pi k t}{T}\right) + b_k \cos\left(\frac{2\pi k t}{T}\right). \quad (1)$$

By increasing the order of the expansion, we can approximate any continuous periodic function $f$ with period $T$ according to the approximation properties of Fourier series. Our temporal embedding can be described as follows:

$$\psi(t) = [\text{ReLU}\left(\text{Linear}\left(\text{Periodic}(t)\right)\right), \text{Trend}(t)],$$

where $t$ is the timestamp, and the two components of the embedding each capture the periodicity and trend of the timestamp. The former part is further defined as

$$\text{Periodic}(t) = [\text{Fourier}(t, T_1), \ldots, \text{Fourier}(t, T_m)],$$

where $T_i$ are $m$ given *periodicity priors* for Fourier-based embedding. In our experiments, we set $T_i$ to represent yearly, monthly, weekly, and daily periodicities. These choices reflect common temporal patterns observed in the datasets. For example, in datasets with natural temporal correlations such as WE, yearly and daily periodicities are effective, while datasets like HI and CT benefit from modeling socially driven patterns captured by monthly and weekly periodicities. Pre-defined periodicity priors yield more stable and interpretable results, as illustrated in Appendix C.2. Each Fourier-based embedding is defined as

$$\text{Fourier}(t, T) = \left[\sin\left(\frac{2\pi k t}{T}\right), \cos\left(\frac{2\pi k t}{T}\right)\right] \in \mathbb{R}^{2K},$$

for orders $k \in \{1, 2, \ldots, K\}$, representing the $K$-term Fourier series expansion. We assign the Fourier orders individually for each periodic prior. By combining multiple periodic components and projecting them through a learnable linear layer, the embedding approximates the aggregation of Fourier coefficients $a_k$ and $b_k$ as formulated in Equation (1). A subsequent ReLU activation promotes sparsity and emphasizes salient frequencies, thereby improving the quality of the learned periodic representations.

In addition to the periodic component, we also provide an optional trend term for the temporal embedding. When the

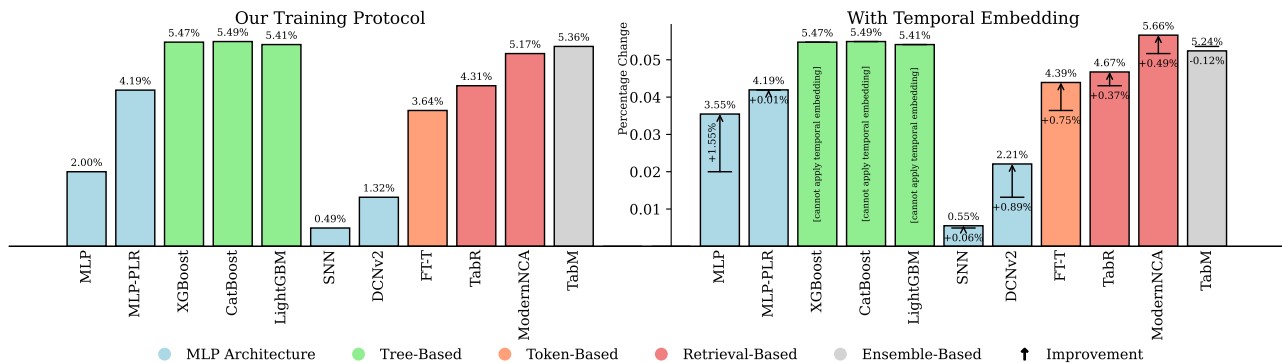

*Figure 7.* **Performance comparison before and after adopting our temporal embedding** into our training protocol on the TabReD benchmark. This figure follows the same setup as Figure 2, allowing for direct comparison. Detailed results are provided in Appendix C.

trend is enabled, the final embedding is augmented with a standardized timestamp, which captures the linear temporal shift beyond the periodic components, represented by

$$\text{Trend}(t) = \text{z-score}(t) \in \mathbb{R}.$$

To thoroughly evaluate the effectiveness of our embedding method, we designed a series of comparative experiments, including the following configurations:

- None: Timestamps are not utilized as the baseline.
- Num: Treating the timestamp as a single numerical feature and directly inputting it into the model.
- Time: Decomposing the timestamp into six numerical features: year, month, day, hour, minute, and second, which introduces partial periodicity information while maintaining a human-readable representation of the timestamp.
- PLR: Applying PLR embedding (Gorishniy et al., 2022) to the timestamp, treating the resulting representations as numerical features to capture temporal patterns adaptively.

We choose MLP, MLP-PLR and ModernNCA for comparison, the results are presented in Table 2. The results indicate that while directly using non-learnable embedding methods can be effective in certain cases, it generally leads to a performance degradation, which is consistent with the common practice of treating timestamps as noise. The performance of PLR embedding is not stable, likely because it discards linear trends and struggles to accurately capture periodic patterns. Results from the MLP-PLR method show no significant improvement, which may be attributed to its incompatibility with the existing numerical embedding.

The models' performance improvement after adopting our temporal embedding is presented in Figure 7. Most methods demonstrate improvements, highlighting the importance of leveraging temporal information in addressing temporal shift tasks. As previously mentioned, MLP-PLR and TabM exhibit limited gains, likely due to their incompatibility with PLR embedding. However, these models achieve substantially greater improvements when the temporal embedding is directly integrated into the model backbone, as shown in Appendix C.2. This suggests that temporal features may

require dedicated embedding strategies rather than relying on existing numerical embedding approaches. After applying temporal embedding, both ModernNCA and TabR demonstrate strong performance, indicating that with an appropriate training protocol and temporal embedding, even retrieval-based methods, which are typically most affected by distributional shift, can regain their practical utility.

The MMD heatmap of the model representation after adopting temporal embedding is shown in Figure 5 right and Appendix C.4. The patterns are closer to the original data, reflecting that it captures correct temporal information, thus effectively alleviating the loss of temporal information during training. The reappearance of diagonal stripes indicates that the model has learned independent representations for each temporal phase within the period, thereby confirming the adaptive role of temporal embedding.

## 7. Conclusion

In this paper, we first investigate the challenges posed by temporal distribution shifts in tabular data, with a focus on effective strategies for addressing them. Starting with a *training protocol* that fully leverages temporal data, we analyze the impact of training lag, validation bias, and the equivalence of validation. Building on these insights, we propose a novel splitting strategy that significantly improves model performance. We further demonstrate that capturing temporal information during training is crucial, and observe that periodic and trend information is often *lost in the learned model representations*. To compensate for this loss, we introduce a *temporal embedding* method that incorporates temporal information from timestamps, improving the model's adaptability to temporal shifts. By combining the new temporal split with the proposed embedding, we observe marked improvements in model performance, particularly for retrieval-based models that previously struggled under temporal shifts. These findings provide valuable insights for advancing deep learning approaches for temporal tabular data, highlighting the importance of both temporal data training protocol and temporal feature integration.

## Acknowledgement

This work is partially supported by National Key R&D Program of China (2024YFE0202800), NSFC (62376118), Key Program of Jiangsu Science Foundation (BK20243012), Collaborative Innovation Center of Novel Software Technology and Industrialization.

## Impact Statement

This paper aims to advance the field of machine learning by addressing the critical challenge of temporal distribution shifts in tabular data, which frequently occur in real-world applications. The proposed temporal training protocol and temporal embedding method offer practical improvements for deploying existing tabular models in open environments.

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

# A. Dataset Explanation

The detailed information about the dataset is provided in Table 3. We adopt the full TabReD dataset (Rubachev et al., 2025) without modification.

| Abbr. | Dataset | Samples | Features | Task Type | Task Description |
|---|---|---|---|---|---|
| HI | Homesite Insurance | 260753 | 296 | Classification | Insurance plan acceptance prediction |
| EO | Ecom Offers | 160057 | 119 | Classification | Predict whether a user will redeem an offer |
| HD | HomeCredit Default | 381664 | 696 | Classification | Loan default prediction |
| SH | Sberbank Housing | 28321 | 387 | Regression | Real estate price prediction |
| CT | Cooking Time | 319986 | 195 | Regression | Restaurant order cooking time estimation |
| DE | Delivery ETA | 350516 | 225 | Regression | Grocery delivery courier ETA prediction |
| MR | Maps Routing | 279945 | 1026 | Regression | Navigation app ETA from live road-graph features |
| WE | Weather | 423795 | 98 | Regression | Weather prediction (temperature) |

*Table 3.* Overview of Datasets. Task descriptions from Rubachev et al. (2025).

In Figure 3, Table 1, and Table 2, we compare the average performance improvement for each method under different strategies (*e.g.* splitting or temporal embedding), specifically the percentage increase in AUC on classification datasets and the percentage decrease in RMSE on regression datasets. In Figure 2 and Figure 7, we present a comparison of performance across different training protocols and methods, where all results are reported as performance improvements relative to the MLP performance under the original split in Rubachev et al. (2025). Since performance improvements between methods can often be influenced by outliers, we apply a robust average, excluding the maximum and minimum performance improvements across the eight datasets before calculating the mean.

The only distinction in the implementation is that TabReD employs different encoding methods during preprocessing for numerical and categorical features for each method-dataset pair. Specifically, it uses identity or noisy-quantile encoding for numerical features and one-hot or ordinal encoding for categorical features. To ensure a fair comparison, we reproduced the experiment from TabReD by removing numerical encoding and fixing categorical encoding to one-hot encoding where necessary. This adjustment is essential for accurately evaluating the performance of the methods and for advancing future research on temporal embeddings.

It is also important to note that the HD dataset suffers from severe class imbalance, which makes it challenging for methods with limited feature extraction capabilities, such as MLP, SNN (Klambauer et al., 2017), and DCNv2 (Wang et al., 2021b), to perform well without additional numerical feature encoding. On this dataset, the AUC of the naive MLP (as well as SNN and DCNv2 methods) drops to approximately 0.55, in contrast to other methods that typically achieve an AUC above 0.80. This large discrepancy renders direct comparison uninformative. To mitigate this issue, we adopt a robust average when reporting the percentage improvement over MLP. Nevertheless, the insights on training protocols and temporal embeddings explored in this work still lead to significant performance improvements on this challenging dataset, as detailed in Appendix C.

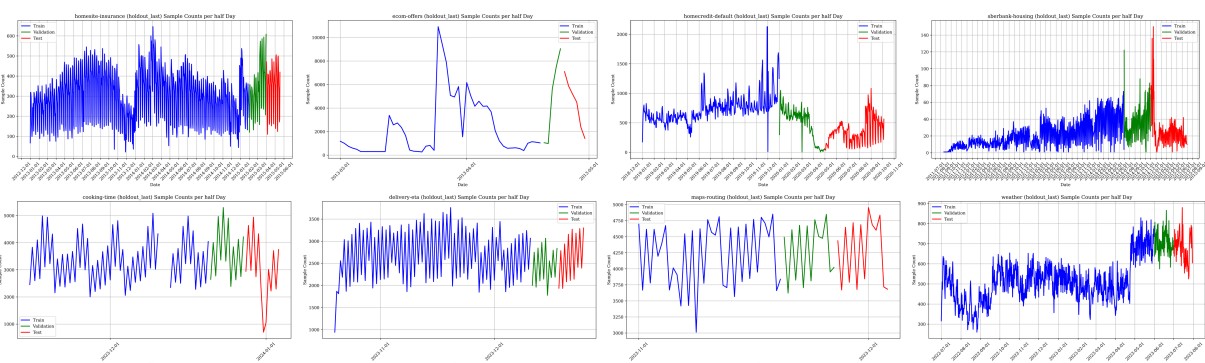

*Figure 8.* The temporal sampling distribution for TabReD datasets. The time slices are divided into half days (*i.e.*, 12 hours).

At the same time, our experiment on the equivalence (Figure 3) of the validation set shows that the datasets whose experimental results seriously fail to meet the equivalence assumption, such as SH and EO, are confirmed to have the most serious nonuniformity in sampling. The temporal sampling distribution for each dataset is shown in Figure 8.

Instead of focusing on the time span between the training and validation sets, we concentrate on ensuring that the number of instances in both sets is equal across different splits, thereby enabling a fair comparison of the splitting strategies. When the temporal sampling distribution becomes highly non-uniform, the observation in Figure 4 no longer holds, as it relies on a constant time span for the validation set. Consequently, our insights regarding the equivalence of the validation sets are significantly affected in such situations. This indicates that the conclusion regarding the equivalence of the validation set on different temporal directions is actually stronger, especially for datasets with uniform sampling over time.

## B. Experimental Setup

Our source code is now available at https://github.com/LAMDA-Tabular/Tabular-Temporal-Shift.

We adopt preprocessing, training, evaluation and tuning setup from Ye et al. (2024) and Liu et al. (2024). We tune hyper-parameters using Optuna (Akiba et al., 2019), performing 100 trials for most methods (except FT-T and TabR for only 25, as explained below) to identify the best configuration. The hyper-parameter search space follows exactly the settings in Rubachev et al. (2025), and can be referred from our source code. The hyper-parameter search space of our proposed temporal embedding is summarized separately in Table 4. Using these optimal hyper-parameters, each method is trained with 15 random seeds, and the average performance across seeds is reported. We label the attribute type (numerical or categorical) for gradient boosting methods such as CatBoost. For all deep learning methods, we use a batch size of 1024 and AdamW (Loshchilov & Hutter, 2019) as the optimizer, with an early stopping patience of 16. For classification tasks, we evaluate models using AUC (higher is better) as the primary metric and use RMSE (lower is better) for regression tasks to select the best-performing model during training on the validation set.

| Parameter | Distribution |
|---|---|
| year_order | 2^UniformInt$[1, 7]$ |
| month_order | 2^UniformInt$[1, 7]$ |
| day_order | 2^UniformInt$[1, 7]$ |
| hour_order | 2^UniformInt$[1, 7]$ |
| trend | $\{$True, False$\}$ |
| d_embedding | 2^UniformInt$[1, 5]$ |

*Table 4.* Temporal embedding hyper-parameter space.

We followed Rubachev et al. (2025) and performed only 25 hyper-parameter tuning trials for FT-T and TabR, as these methods exhibit lower efficiency on datasets with large feature dimensions and sample sizes. For our temporal embedding, we conducted separate hyper-parameter searches for the periodic order and linear trend, to selectively extract temporal information while reducing computational overhead. However, since 25 tuning trials were insufficient to identify the optimal hyper-parameters for the temporal embedding, we performed a global tuning of the temporal embedding order for FT-T and TabR, with all periodic components sharing the same order.

To ensure the validity of random splitting, each group of random split experiments was tested on three distinct random splits, with 15 random seeds run on each split. The mean performance across these runs is reported as the final result. The standard deviation of the random split is calculated based on all 45 results (3 splits × 15 seeds), as the random split is subject to variability from both the split selection and the running seeds during the training phase. This approach better reflects the overall stability of the standard procedure.

# C. Additional Results

## C.1. Splitting Strategies

| Methods | Splits | HI↑ | EO↑ | HD↑ | SH↓ | CT↓ | DE↓ | MR↓ | WE↓ | Avg. Imp. |
|---|---|---|---|---|---|---|---|---|---|---|
| Mambular | Original | 0.6782 | 0.5713 | 0.5872 | 0.6822 | 0.5107 | 0.5981 | 0.2330 | 2.2232 | – |
| | Ours | 0.6870 | 0.6010 | 0.5396 | 0.5492 | 0.5080 | 0.5999 | 0.2297 | 2.1972 | +2.59% |
| TabPFN v2 | Original | 0.8215 | 0.5660 | 0.5000 | 0.2276 | 0.4858 | 0.5529 | 0.1673 | 1.7076 | – |
| | Ours | 0.8205 | 0.5717 | 0.5000 | 0.2204 | 0.4804 | 0.5619 | 0.1740 | 1.6031 | +0.71% |

*Table 5.* The performance comparison of the autoregressive method Mambular (Thielmann et al., 2024) and the tabular general method TabPFN v2 (Hollmann et al., 2025) under original temporal split in Rubachev et al. (2025) and our refined temporal split. Despite the large dataset size leading to suboptimal overall performance, Mambular benefits significantly from our refined temporal split. For TabPFN v2, since no training is required, we modified the context selection: 10,000 contexts were randomly chosen (Original) and the last 10,000 samples were selected as the context (Ours). The results also show a performance improvement.

To further validate the general applicability of our training protocol, we apply our temporal splitting strategy to two new model families: the autoregressive model Mambular (Thielmann et al., 2024) and the tabular general model TabPFN-v2 (Hollmann et al., 2025). As shown in Table 5, our temporal split consistently improves performance compared to the original splits used by these models.

We further report detailed experimental results for the different data splitting strategies illustrated in Figure 3, with corresponding metrics presented in Table 6. Specifically, Table 7 and Table 8 compare the performance and standard deviation across three strategies: the baseline temporal split from Rubachev et al. (2025), the random split discussed in Section 1, and our proposed temporal split described in Section 4.4. These results demonstrate the superior performance and stability of our refined temporal split. Furthermore, Figure 9 presents an additional experiment using the robustness score to further highlight the consistency of our splitting strategy.

| Methods | Splits | HI↑ | EO↑ | HD↑ | SH↓ | CT↓ | DE↓ | MR↓ | WE↓ | Avg. Imp. |
|---|---|---|---|---|---|---|---|---|---|---|
| MLP-PLR | (c) | 0.9484 | 0.5630 | 0.8426 | 0.2635 | 0.4828 | 0.5515 | 0.1629 | 1.5693 | – |
| | (a) | 0.9575 | 0.5765 | 0.8470 | 0.2629 | 0.4826 | 0.5545 | 0.1630 | 1.5623 | +0.51% |
| | (b) | 0.9590 | 0.5928 | 0.8481 | 0.2375 | 0.4811 | 0.5539 | 0.1630 | 1.5713 | +2.09% |
| | (d) | 0.9476 | 0.5654 | 0.8451 | 0.2460 | 0.4814 | 0.5490 | 0.1637 | 1.5553 | +1.06% |
| ModernNCA | (c) | 0.9469 | 0.5648 | 0.8424 | 0.2846 | 0.4846 | 0.5503 | 0.1637 | 1.5200 | – |
| | (a) | 0.9564 | 0.5731 | 0.8444 | 0.2836 | 0.4832 | 0.5520 | 0.1634 | 1.5165 | +0.44% |
| | (b) | 0.9603 | 0.5898 | 0.8469 | 0.2430 | 0.4813 | 0.5514 | 0.1639 | 1.5258 | +2.63% |
| | (d) | 0.9539 | 0.5615 | 0.8405 | 0.2469 | 0.4824 | 0.5533 | 0.1636 | 1.5391 | +1.49% |
| TabM | (c) | 0.9543 | 0.5756 | 0.8530 | 0.2725 | 0.4821 | 0.5502 | 0.1629 | 1.5210 | – |
| | (a) | 0.9580 | 0.5809 | 0.8520 | 0.2587 | 0.4816 | 0.5546 | 0.1626 | 1.5080 | +0.83% |
| | (b) | 0.9622 | 0.5996 | 0.8531 | 0.2448 | 0.4796 | 0.5543 | 0.1620 | 1.5263 | +1.90% |
| | (d) | 0.9586 | 0.5772 | 0.8464 | 0.2457 | 0.4818 | 0.5482 | 0.1626 | 1.5165 | +1.34% |

*Table 6.* Detailed performance results (AUC for classification and RMSE for regression) of different splitting strategies illustrated in Figure 3 left. These results are plotted in Figure 3 right.

| Methods | Splits | HI↑ | EO↑ | HD↑ | SH↓ | CT↓ | DE↓ | MR↓ | WE↓ | Avg. Imp. |
|---|---|---|---|---|---|---|---|---|---|---|
| MLP | Original | 0.9404 | 0.5866 | 0.4730 | 0.2802 | 0.4820 | 0.5526 | 0.1624 | 1.5331 | – |
| | Random | 0.9383 | 0.6225 | 0.5532 | 0.2509 | 0.4814 | 0.5521 | 0.1619 | 1.5252 | +4.30% |
| | Ours | 0.9360 | 0.6220 | 0.5508 | 0.2641 | 0.4821 | 0.5515 | 0.1619 | 1.5362 | +3.50% |
| MLP-PLR | Original | 0.9592 | 0.5816 | 0.8448 | 0.2412 | 0.4811 | 0.5533 | 0.1616 | 1.5185 | – |
| | Random | 0.9599 | 0.6225 | 0.8208 | 0.2406 | 0.4800 | 0.5507 | 0.1616 | 1.5097 | +0.73% |
| | Ours | 0.9596 | 0.6185 | 0.8166 | 0.2361 | 0.4799 | 0.5481 | 0.1616 | 1.5235 | +0.75% |
| FT-T | Original | 0.9562 | 0.5791 | 0.5301 | 0.2600 | 0.4814 | 0.5534 | 0.1627 | 1.5155 | – |
| | Random | 0.9616 | 0.6268 | 0.5846 | 0.2369 | 0.4804 | 0.5503 | 0.1622 | 1.5001 | +3.76% |
| | Ours | 0.9591 | 0.6159 | 0.5746 | 0.2438 | 0.4807 | 0.5503 | 0.1623 | 1.5146 | +2.78% |
| SNN | Original | 0.9538 | 0.5795 | 0.5266 | 0.3292 | 0.4834 | 0.5543 | 0.1656 | 1.5604 | – |
| | Random | 0.9535 | 0.6187 | 0.5358 | 0.3228 | 0.4829 | 0.5543 | 0.1648 | 1.5600 | +1.38% |
| | Ours | 0.9480 | 0.6209 | 0.5591 | 0.3367 | 0.4825 | 0.5541 | 0.1648 | 1.5611 | +1.38% |
| DCNv2 | Original | 0.9519 | 0.5846 | 0.5082 | 0.3425 | 0.4827 | 0.5519 | 0.1628 | 1.5305 | – |
| | Random | 0.9486 | 0.6195 | 0.5484 | 0.3340 | 0.4814 | 0.5516 | 0.1623 | 1.5264 | +2.11% |
| | Ours | 0.9460 | 0.6244 | 0.5505 | 0.3129 | 0.4824 | 0.5521 | 0.1622 | 1.5216 | +3.01% |
| TabR | Original | 0.9527 | 0.5727 | 0.8442 | 0.2676 | 0.4818 | 0.5557 | 0.1625 | 1.4782 | – |
| | Random | 0.9543 | 0.6206 | 0.8147 | 0.2384 | 0.4880 | 0.5548 | 0.1622 | 1.4629 | +2.00% |
| | Ours | 0.9605 | 0.6148 | 0.8342 | 0.2370 | 0.4883 | 0.5550 | 0.1623 | 1.4732 | +2.20% |
| ModernNCA | Original | 0.9571 | 0.5712 | 0.8487 | 0.2526 | 0.4817 | 0.5523 | 0.1631 | 1.4977 | – |
| | Random | 0.9617 | 0.6246 | 0.8399 | 0.2299 | 0.4806 | 0.5510 | 0.1621 | 1.4773 | +2.53% |
| | Ours | 0.9610 | 0.6341 | 0.8378 | 0.2325 | 0.4804 | 0.5520 | 0.1619 | 1.4857 | +2.49% |
| TabM | Original | 0.9590 | 0.5952 | 0.8549 | 0.2465 | 0.4799 | 0.5522 | 0.1610 | 1.4852 | – |
| | Random | 0.9629 | 0.6332 | 0.8282 | 0.2305 | 0.4794 | 0.5495 | 0.1607 | 1.4681 | +1.51% |
| | Ours | 0.9640 | 0.6325 | 0.8290 | 0.2306 | 0.4813 | 0.5500 | 0.1612 | 1.4887 | +1.25% |
| Linear | Original | 0.9388 | 0.5731 | 0.8174 | 0.2560 | 0.4879 | 0.5587 | 0.1744 | 1.7465 | – |
| | Random | 0.9397 | 0.5895 | 0.8235 | 0.2458 | 0.4867 | 0.5591 | 0.1685 | 1.7425 | +1.44% |
| | Ours | 0.9388 | 0.5944 | 0.8231 | 0.2435 | 0.4864 | 0.5596 | 0.1680 | 1.7464 | +1.64% |
| XGBoost | Original | 0.9609 | 0.5764 | 0.8627 | 0.2475 | 0.4823 | 0.5459 | 0.1616 | 1.4699 | – |
| | Random | 0.9625 | 0.6200 | 0.8452 | 0.2298 | 0.4806 | 0.5468 | 0.1611 | 1.4566 | +1.79% |
| | Ours | 0.9625 | 0.6199 | 0.8644 | 0.2262 | 0.4792 | 0.5520 | 0.1610 | 1.4700 | +2.06% |
| CatBoost | Original | 0.9612 | 0.5671 | 0.8588 | 0.2469 | 0.4824 | 0.5464 | 0.1619 | 1.4715 | – |
| | Random | 0.9639 | 0.6213 | 0.8580 | 0.2340 | 0.4805 | 0.5471 | 0.1613 | 1.4556 | +2.09% |
| | Ours | 0.9639 | 0.6242 | 0.8620 | 0.2292 | 0.4792 | 0.5495 | 0.1610 | 1.4654 | +2.37% |
| LightGBM | Original | 0.9600 | 0.5633 | 0.8580 | 0.2452 | 0.4826 | 0.5474 | 0.1618 | 1.4723 | – |
| | Random | 0.9616 | 0.6136 | 0.8334 | 0.2322 | 0.4807 | 0.5469 | 0.1616 | 1.4471 | +1.73% |
| | Ours | 0.9631 | 0.6164 | 0.8599 | 0.2260 | 0.4877 | 0.5500 | 0.1612 | 1.4654 | +2.14% |
| RandomForest | Original | 0.9537 | 0.5755 | 0.7971 | 0.2623 | 0.4870 | 0.5565 | 0.1653 | 1.5839 | – |
| | Random | 0.9580 | 0.6254 | 0.8142 | 0.2427 | 0.4846 | 0.5588 | 0.1649 | 1.5694 | +2.50% |
| | Ours | 0.9580 | 0.6068 | 0.8171 | 0.2400 | 0.4841 | 0.5588 | 0.1647 | 1.5845 | +2.17% |

*Table 7.* Detailed performance results (AUC for classification and RMSE for regression) of different splitting strategies on the TabReD benchmark: the original temporal split from Rubachev et al. (2025), the random split in Section 1, and our proposed temporal split in Section 4.4. These results are plotted in Figure 2 left, Figure 2 right, and Figure 7 left, respectively, while also illustrated in Table 1.

| Methods | Splits | HI↑ | EO↑ | HD↑ | SH↓ | CT↓ | DE↓ | MR↓ | WE↓ | Avg. Imp. |
|---|---|---|---|---|---|---|---|---|---|---|
| MLP | Original | 0.0026 | 0.0033 | 0.0006 | 0.0281 | 0.0005 | 0.0012 | 0.0001 | 0.0050 | – |
| | Random | 0.0042 | 0.0031 | 0.0011 | 0.0119 | 0.0004 | 0.0006 | 0.0001 | 0.0059 | +1.81% |
| | Ours | 0.0053 | 0.0040 | 0.0015 | 0.0093 | 0.0006 | 0.0012 | 0.0001 | 0.0059 | +29.68% |
| MLP-PLR | Original | 0.0005 | 0.0035 | 0.0028 | 0.0043 | 0.0004 | 0.0014 | 0.0001 | 0.0046 | – |
| | Random | 0.0014 | 0.0055 | 0.0102 | 0.0131 | 0.0003 | 0.0010 | 0.0002 | 0.0163 | +125.87% |
| | Ours | 0.0004 | 0.0072 | 0.0084 | 0.0026 | 0.0003 | 0.0011 | 0.0003 | 0.0038 | +50.91% |
| FT-T | Original | 0.0092 | 0.0061 | 0.0278 | 0.0142 | 0.0004 | 0.0028 | 0.0005 | 0.0058 | – |
| | Random | 0.0036 | 0.0095 | 0.0391 | 0.0082 | 0.0007 | 0.0014 | 0.0002 | 0.0084 | +0.15% |
| | Ours | 0.0048 | 0.0129 | 0.0319 | 0.0119 | 0.0008 | 0.0011 | 0.0003 | 0.0051 | +5.59% |
| SNN | Original | 0.0006 | 0.0109 | 0.0166 | 0.0709 | 0.0009 | 0.0029 | 0.0003 | 0.0034 | – |
| | Random | 0.0014 | 0.0053 | 0.0065 | 0.0776 | 0.0009 | 0.0023 | 0.0005 | 0.0132 | +43.95% |
| | Ours | 0.0011 | 0.0067 | 0.0154 | 0.1038 | 0.0007 | 0.0020 | 0.0005 | 0.0054 | +16.77% |
| DCNv2 | Original | 0.0013 | 0.0061 | 0.0239 | 0.0810 | 0.0003 | 0.0006 | 0.0002 | 0.0068 | – |
| | Random | 0.0020 | 0.0066 | 0.0160 | 0.0660 | 0.0004 | 0.0005 | 0.0001 | 0.0094 | +0.06% |
| | Ours | 0.0019 | 0.0060 | 0.0158 | 0.0658 | 0.0007 | 0.0009 | 0.0001 | 0.0040 | +8.86% |
| TabR | Original | 0.0024 | 0.0065 | 0.0041 | 0.0170 | 0.0003 | 0.0016 | 0.0005 | 0.0062 | – |
| | Random | 0.0052 | 0.0055 | 0.0165 | 0.0068 | 0.0037 | 0.0033 | 0.0003 | 0.0067 | +224.06% |
| | Ours | 0.0009 | 0.0095 | 0.0044 | 0.0045 | 0.0017 | 0.0040 | 0.0004 | 0.0076 | +82.17% |
| ModernNCA | Original | 0.0060 | 0.0037 | 0.0018 | 0.0089 | 0.0007 | 0.0018 | 0.0001 | 0.0052 | – |
| | Random | 0.0007 | 0.0110 | 0.0074 | 0.0037 | 0.0005 | 0.0015 | 0.0004 | 0.0101 | +74.89% |
| | Ours | 0.0009 | 0.0030 | 0.0059 | 0.0033 | 0.0005 | 0.0007 | 0.0001 | 0.0034 | −10.79% |
| TabM | Original | 0.0018 | 0.0057 | 0.0024 | 0.0092 | 0.0007 | 0.0014 | 0.0001 | 0.0046 | – |
| | Random | 0.0014 | 0.0030 | 0.0128 | 0.0033 | 0.0007 | 0.0013 | 0.0002 | 0.0047 | +44.34% |
| | Ours | 0.0002 | 0.0042 | 0.0080 | 0.0030 | 0.0010 | 0.0014 | 0.0004 | 0.0049 | +37.66% |
| Linear | Original | 0.0005 | 0.0058 | 0.0008 | 0.0134 | 0.0004 | 0.0008 | 0.0107 | 0.0031 | – |
| | Random | 0.0009 | 0.0040 | 0.0018 | 0.0072 | 0.0003 | 0.0004 | 0.0073 | 0.0029 | +4.96% |
| | Ours | 0.0009 | 0.0165 | 0.0006 | 0.0007 | 0.0002 | 0.0008 | 0.0008 | 0.0014 | −8.59% |
| XGBoost | Original | 0.0002 | 0.0005 | 0.0005 | 0.0004 | 0.0001 | 0.0001 | 0.0000 | 0.0008 | – |
| | Random | 0.0003 | 0.0024 | 0.0091 | 0.0016 | 0.0002 | 0.0002 | 0.0001 | 0.0064 | +456.05% |
| | Ours | 0.0001 | 0.0005 | 0.0002 | 0.0002 | 0.0001 | 0.0000 | 0.0001 | 0.0011 | −15.20% |
| CatBoost | Original | 0.0003 | 0.0110 | 0.0005 | 0.0014 | 0.0001 | 0.0002 | 0.0000 | 0.0016 | – |
| | Random | 0.0008 | 0.0025 | 0.0022 | 0.0017 | 0.0003 | 0.0002 | 0.0001 | 0.0021 | +104.61% |
| | Ours | 0.0002 | 0.0018 | 0.0004 | 0.0007 | 0.0001 | 0.0001 | 0.0000 | 0.0011 | −30.61% |
| LightGBM | Original | 0.0002 | 0.0008 | 0.0004 | 0.0004 | 0.0001 | 0.0002 | 0.0000 | 0.0013 | – |
| | Random | 0.0005 | 0.0045 | 0.0081 | 0.0015 | 0.0003 | 0.0003 | 0.0004 | 0.0042 | +616.12% |
| | Ours | 0.0001 | 0.0008 | 0.0006 | 0.0002 | 0.0002 | 0.0001 | 0.0000 | 0.0012 | +8.59% |
| RandomForest | Original | 0.0001 | 0.0008 | 0.0008 | 0.0006 | 0.0001 | 0.0001 | 0.0000 | 0.0004 | – |
| | Random | 0.0002 | 0.0029 | 0.0017 | 0.0024 | 0.0001 | 0.0003 | 0.0000 | 0.0004 | +134.95% |
| | Ours | 0.0001 | 0.0004 | 0.0007 | 0.0004 | 0.0001 | 0.0001 | 0.0000 | 0.0004 | −7.65% |

*Table 8.* Detailed standard deviation of performance results (AUC for classification and RMSE for regression) when adopting different splitting strategies on the TabReD benchmark: the original temporal split from Rubachev et al. (2025), the random split in Section 1, and our proposed temporal split in Section 4.4. These results are illustrated in Table 1.

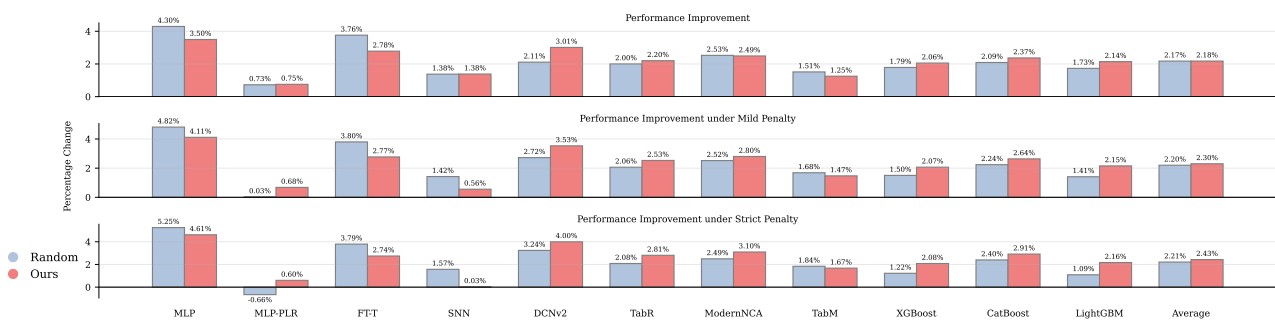

*Figure 9.* Comparison of model performance and stability of our splitting strategy and random split using robustness score. From top to bottom, the figure illustrates the mean performance improvement over baseline temporal split adopted in Rubachev et al. (2025) under no penalty, mild penalty, and strict penalty, where the first panel demonstrates the performance, while the latter two highlight the robustness.

The thorough comparison of the original temporal split in (Rubachev et al., 2025), the random split in Section 1, and our newly proposed temporal split in Section 4.4 is demonstrated in Figure 9. Our proposed splitting strategy results in a comparable performance to random split while offering better stability. Since random split suffers a severe instability shown in Table 1, including the splitting selection and the running seed, which cannot be estimated at the training stage, we use robustness score for comparison, which is defined as

$$RS_k = \mu - k\sigma,$$

where $\mu$, $\sigma$ are the average and the standard deviation of model performance. We employed *no penalty*, *mild penalty*, and *strict penalty*, corresponding to $k = 0$, $k = 1$, and $k = 2$, respectively. We observed that without penalty, our temporal split performs comparably to the random split (2.18% *vs.* 2.17% on average), and as the penalty increases, the advantage of our splitting strategy becomes more pronounced (2.43% *vs.* 2.21% on average). This indicates that our method not only achieves strong performance but also exhibits superior robustness.

### C.2. Temporal Embeddings

We conducted a further analysis on the incompatibility between temporal embedding and PLR embedding discussed in Section 6. Our temporal embedding converts timestamps into numerical inputs, which may be incompatible with PLR numerical embedding (Gorishniy et al., 2022). Specifically, once timestamps are embedded, their representation reflects temporal similarity. Applying another periodic transformation via PLR could increase optimization difficulty. Directly feed the temporal embedding into the model backbone consistently improves, as demonstrated in Table 9.

| Methods | Emb. | HI↑ | EO↑ | HD↑ | SH↓ | CT↓ | DE↓ | MR↓ | WE↓ | Avg. Imp. |
|---|---|---|---|---|---|---|---|---|---|---|
| | None | 0.9596 | 0.6185 | 0.8166 | 0.2361 | 0.4799 | 0.5481 | 0.1616 | 1.5235 | – |
| MLP-PLR | +Ours | 0.9607 | 0.6110 | 0.8158 | 0.2338 | 0.4803 | 0.5494 | 0.1617 | 1.5133 | +0.01% |
| | +Ours* | 0.9609 | 0.6215 | 0.8155 | 0.2337 | 0.4788 | 0.5492 | 0.1616 | 1.5152 | **+0.26%** |
| | None | 0.9640 | 0.6325 | 0.8290 | 0.2306 | 0.4813 | 0.5500 | 0.1612 | 1.4887 | – |
| TabM | +Ours | 0.9629 | 0.6271 | 0.8363 | 0.2321 | 0.4791 | 0.5488 | 0.1609 | 1.4812 | +0.07% |
| | +Ours* | 0.9633 | 0.6260 | 0.8368 | 0.2297 | 0.4775 | 0.5490 | 0.1607 | 1.4873 | **+0.20%** |
| | None | 0.9610 | 0.6341 | 0.8378 | 0.2325 | 0.4804 | 0.5520 | 0.1619 | 1.4857 | – |
| ModernNCA | +Ours | 0.9620 | 0.6356 | 0.8316 | 0.2255 | 0.4791 | 0.5535 | 0.1617 | 1.4903 | +0.30% |
| | +Ours* | 0.9623 | 0.6357 | 0.8344 | 0.2262 | 0.4790 | 0.5515 | 0.1616 | 1.4855 | **+0.41%** |

* Feed temporal embedding directly to the backbone.

*Table 9.* The performance comparison between the temporal embedding used in our paper and directly feeding the temporal encoding into the model backbone. All three methods show improvement, indicating that there may be incompatibility between the temporal embedding and numerical embedding.

We also experimented with tuning hyperparameters to find the cycles, results in Table 10. When using fixed prior periods, ModernNCA achieved a 0.30% performance improvement, while setting adjustable cycles resulted in a -2.48% performance decline. In temporal distribution shift scenarios, due to the absence of an entirely accurate validation set, we believe that prior knowledge of fixed cycles is more stable and interpretable than adjustable cycles. It's also important to note that in many tasks, complete cycles are not available. For example, in the WE dataset, there is a yearly cycle, but the training set does not span a full year, which highlights the importance of prior knowledge.

| Methods | Cycles | HI↑ | EO↑ | HD↑ | SH↓ | CT↓ | DE↓ | MR↓ | WE↓ | Avg. Imp. |
|---------|--------|-----|-----|-----|-----|-----|-----|-----|-----|-----------|
| | None | 0.9610 | 0.6341 | 0.8378 | 0.2325 | 0.4804 | 0.5520 | 0.1619 | 1.4857 | – |
| ModernNCA | Fixed | 0.9620 | 0.6356 | 0.8316 | 0.2255 | 0.4791 | 0.5535 | 0.1617 | 1.4903 | **+0.30%** |
| | Adjustable | 0.9575 | 0.5719 | 0.8355 | 0.2469 | 0.4820 | 0.5493 | 0.1630 | 1.4926 | −2.20% |

*Table 10.* When using adjustable cycles, the performance comparison with no temporal information (none) and our temporal embedding (fixed) shows that ModernNCA experiences a performance drop of -2.48%, trailing behind the fixed cycle temporal embedding (+0.30%). This highlights that, in temporal shift scenarios, tuning cycles based on the validation set is less reliable than using fixed prior cycles.

We further present the detailed performance results of different embedding methods discussed in Section 6 in Table 11, as well as the model performance before and after adopting our proposed temporal embedding in Table 12.

| Methods | Emb. | HI↑ | EO↑ | HD↑ | SH↓ | CT↓ | DE↓ | MR↓ | WE↓ | Avg. Imp. |
|---------|------|-----|-----|-----|-----|-----|-----|-----|-----|-----------|
| | None | 0.9360 | 0.6220 | 0.5508 | 0.2641 | 0.4821 | 0.5515 | 0.1619 | 1.5362 | – |
| | +Num | 0.9350 | 0.6233 | 0.5505 | 0.2594 | 0.4802 | 0.5651 | 0.1617 | 1.5394 | −0.04% |
| MLP | +Time | 0.9353 | 0.6241 | 0.5514 | 0.2435 | 0.5407 | 0.5640 | 0.1619 | 1.5272 | −0.70% |
| | +PLR | 0.9482 | 0.6186 | 0.5509 | 0.2513 | 0.4814 | 0.5517 | 0.1621 | 1.5361 | +0.70% |
| | +Ours | 0.9471 | 0.6252 | 0.5519 | 0.2431 | 0.4801 | 0.5518 | 0.1621 | 1.5319 | +1.31% |
| | None | 0.9596 | 0.6185 | 0.8166 | 0.2361 | 0.4799 | 0.5481 | 0.1616 | 1.5235 | – |
| | +Num | 0.9594 | 0.6248 | 0.8088 | 0.2353 | 0.4807 | 0.5524 | 0.1611 | 1.5270 | −0.06% |
| MLP-PLR | +Time | 0.9593 | 0.6185 | 0.8303 | 0.2347 | 0.4811 | 0.5645 | 0.1615 | 1.5275 | −0.15% |
| | +PLR | 0.9595 | 0.6179 | 0.8193 | 0.2331 | 0.4829 | 0.5536 | 0.1614 | 1.5212 | +0.01% |
| | +Ours | 0.9607 | 0.6110 | 0.8158 | 0.2338 | 0.4803 | 0.5494 | 0.1617 | 1.5133 | +0.01% |
| | None | 0.9610 | 0.6341 | 0.8378 | 0.2325 | 0.4804 | 0.5520 | 0.1619 | 1.4857 | – |
| | +Num | 0.9616 | 0.6354 | 0.8267 | 0.2268 | 0.4804 | 0.5575 | 0.1630 | 1.4862 | −0.04% |
| ModernNCA | +Time | 0.9614 | 0.6317 | 0.8384 | 0.2304 | 0.4819 | 0.5665 | 0.1625 | 1.4841 | −0.32% |
| | +PLR | 0.9608 | 0.6334 | 0.8366 | 0.2299 | 0.4775 | 0.5599 | 0.1618 | 1.4854 | +0.02% |
| | +Ours | 0.9620 | 0.6356 | 0.8316 | 0.2255 | 0.4791 | 0.5535 | 0.1617 | 1.4903 | +0.30% |

*Table 11.* Detailed performance results (AUC for classification and RMSE for regression) of different embedding methods discussed in Section 6 within our training protocol on the TabReD benchmark. The average improvements are illustrated in Table 2.

| Methods | Emb. | HI↑ | EO↑ | HD↑ | SH↓ | CT↓ | DE↓ | MR↓ | WE↓ | Avg. Imp. |
|---|---|---|---|---|---|---|---|---|---|---|
| MLP | None | 0.9360 | 0.6220 | 0.5508 | 0.2641 | 0.4821 | 0.5515 | 0.1619 | 1.5362 | – |
| | +Ours | 0.9471 | 0.6252 | 0.5519 | 0.2431 | 0.4801 | 0.5518 | 0.1621 | 1.5319 | +1.31% |
| MLP-PLR | None | 0.9596 | 0.6185 | 0.8166 | 0.2361 | 0.4799 | 0.5481 | 0.1616 | 1.5235 | – |
| | +Ours | 0.9607 | 0.6110 | 0.8158 | 0.2338 | 0.4803 | 0.5494 | 0.1617 | 1.5133 | +0.01% |
| FT-T | None | 0.9591 | 0.6159 | 0.5746 | 0.2438 | 0.4807 | 0.5503 | 0.1623 | 1.5146 | – |
| | +Ours | 0.9608 | 0.6211 | 0.5563 | 0.2363 | 0.4778 | 0.5503 | 0.1622 | 1.5118 | +0.22% |
| SNN | None | 0.9480 | 0.6209 | 0.5591 | 0.3367 | 0.4825 | 0.5541 | 0.1648 | 1.5611 | – |
| | +Ours | 0.9484 | 0.6232 | 0.5547 | 0.2865 | 0.4824 | 0.5551 | 0.1648 | 1.5605 | +1.81% |
| DCNv2 | None | 0.9460 | 0.6244 | 0.5505 | 0.3129 | 0.4824 | 0.5521 | 0.1622 | 1.5216 | – |
| | +Ours | 0.9454 | 0.6196 | 0.5388 | 0.2629 | 0.4809 | 0.5516 | 0.1623 | 1.5246 | +1.64% |
| TabR | None | 0.9605 | 0.6148 | 0.8342 | 0.2370 | 0.4883 | 0.5550 | 0.1623 | 1.4732 | – |
| | +Ours | 0.9606 | 0.6233 | 0.8426 | 0.2392 | 0.4827 | 0.5497 | 0.1627 | 1.4620 | +0.52% |
| ModernNCA | None | 0.9610 | 0.6341 | 0.8378 | 0.2325 | 0.4804 | 0.5520 | 0.1619 | 1.4857 | – |
| | +Ours | 0.9620 | 0.6356 | 0.8316 | 0.2255 | 0.4791 | 0.5535 | 0.1617 | 1.4903 | +0.30% |
| TabM | None | 0.9640 | 0.6325 | 0.8290 | 0.2306 | 0.4813 | 0.5500 | 0.1612 | 1.4887 | – |
| | +Ours | 0.9629 | 0.6271 | 0.8363 | 0.2321 | 0.4791 | 0.5488 | 0.1609 | 1.4812 | +0.07% |

*Table 12.* Detailed performance results (AUC for classification and RMSE for regression) before and after adopting the proposed temporal embedding method from Section 6 within our training protocol on the TabReD benchmark. These results are plotted in Figure 7 right. All methods demonstrated improvement after using our temporal embedding, with an average performance improvement of 0.74%.

### C.3. Performance Rankings

We present the performance rankings of the model under different splits, as well as the rankings when temporal embedding is adopted to our training protocol, in Table 13. The results indicate that TabM, CatBoost, and XGBoost consistently perform well, while ModernNCA excels under the random split and our training protocol, and MLP-PLR performs the opposite.

It is worth noting that although the relative improvement of TabM over MLP decreases after adding the temporal embedding (−0.12% in Figure 7), TabM itself still achieves a 0.07% performance gain (shown in Table 12) and rises by 0.875 ranks in the average model ranking (shown in Table 13). Notably, none of the other methods experience a performance drop. This provides a comprehensive multi-perspective evaluation.

| Splits | MLP | PLR | FT-T | SNN | DCNv2 | TabR | MNCA | TabM | XGBoost | CatBoost | LGBM |
|---|---|---|---|---|---|---|---|---|---|---|---|
| Original | 7.750 | 4.375 | 6.875 | 9.375 | 8.250 | 7.375 | 6.500 | **3.125** | 3.375 | 4.250 | 4.750 |
| Random | 8.250 | 5.625 | 5.625 | 10.250 | 9.625 | 8.000 | 4.750 | **2.750** | 3.125 | 3.125 | 4.875 |
| Ours | 8.000 | 5.750 | 7.500 | 9.500 | 8.375 | 8.125 | 4.875 | 4.000 | 3.375 | **2.125** | 4.375 |
| Ours +temporal embedding | 7.875 | 6.625 | 6.250 | 9.625 | 9.250 | 6.250 | 4.625 | 3.125 | 4.500 | **2.875** | 5.000 |

*Table 13.* Performance rankings of original temporal split in (Rubachev et al., 2025), random split in Figure 2, and our proposed temporal split in Section 4.4 with and without our temporal embedding, measured by the *average performance ranking* on the TabReD benchmark, as an extension of Table 1. "PLR," "MNCA," and "LGBM" denote "MLP-PLR," "ModernNCA," and "LightGBM," respectively.

## C.4. Model Representations

We present the model representations of TabM before and after incorporating temporal embedding in Figure 10, as a representative SOTA model complementary to the MLP shown in Figure 5. Since our TabM internally ensembles 32 model representations, we compute their average to reduce computational cost, which aligns with how the model operates during inference. The left plot exhibits a clear grid structure and shows richer representations than MLP, reflecting the stronger representational capacity of TabM. However, the absence of diagonal stripes suggests that the model still primarily captures only two coarse temporal modes (*i.e.*, weekdays and weekends). In contrast, the right plot shows the model representation after incorporating temporal embedding, where both the grid structure and diagonal patterns are clearly visible, indicating that the model has learned more fine-grained temporal knowledge.

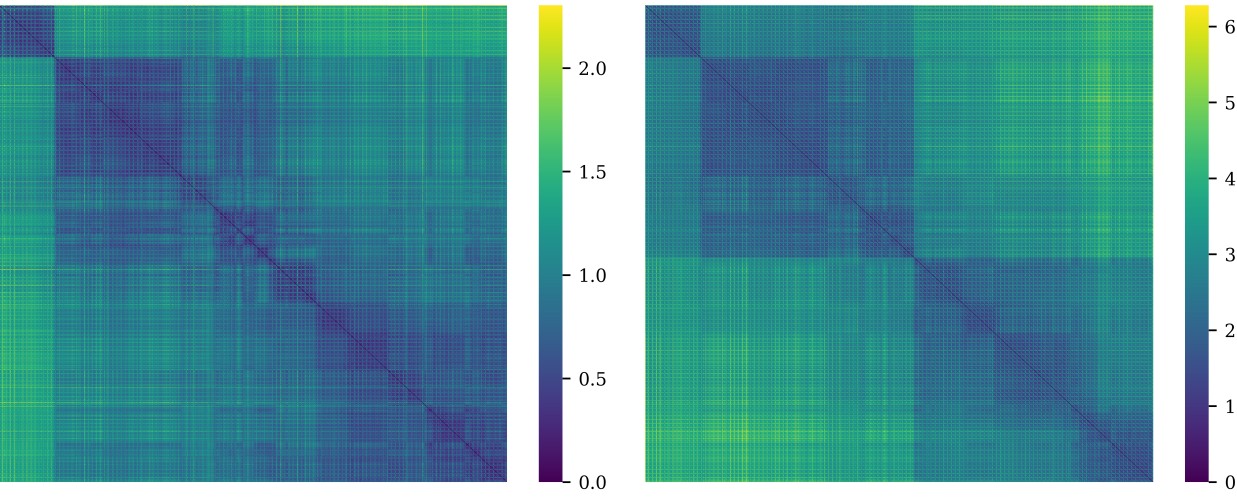

*Figure 10.* MMD heatmaps of the model representations of TabM before (left) and after (right) incorporating temporal embedding. The visualizations are obtained by averaging the 32 internal representations of the ensemble. The observed patterns are consistent with those shown in Figure 5.

