# OpenReview forum: "Understanding the Limits of Deep Tabular Methods with Temporal Shift"
_ICML.cc/2025/Conference — ICML 2025 poster_

### Official Review · Reviewer_nBxA · 2025-03-12

**Overall Recommendation:** 3

**Summary:**

- The paper analyses temporal splits for tabular DL. It proposes a new splitting strategy and also analyzes how random splitting affects performance. Additionally, the authors propose temporal embeddings, using fourier transformation and somewhat following the ideas proposed in [1] with PLR embeddings


---
[1] Gorishniy, Yury, Ivan Rubachev, and Artem Babenko. "On embeddings for numerical features in tabular deep learning." Advances in Neural Information Processing Systems 35 (2022): 24991-25004.

**Claims And Evidence:**

-  The claim that the proposed new temporal split is superior is not supported by the results (Figure 5) as the random split seems to perform nearly identical
   - While that claim is not met, the finding that the random split outperforms the split from Rubachev (2025) is an interesting finding itself. If the claims in the abstract/contributions that the newly proposed method "offers substantial improvements" are scaled down a bit, this is not an issue
- The proposed temporal embedding does also not offer any improvement over the known PLR embeddings (Figure 8)

**Essential References Not Discussed:**

- I wonder how these splits effect ICL models (TabICL, TabPFN v2) [1, 2].
- Other than that all necessary work is included and adequately adressed.


---
[1] Qu, Jingang, et al. "TabICL: A Tabular Foundation Model for In-Context Learning on Large Data." arXiv preprint arXiv:2502.05564 (2025).
[2] Hollmann, Noah, et al. "Accurate predictions on small data with a tabular foundation model." Nature 637.8045 (2025): 319-326.

**Experimental Designs Or Analyses:**

- The experimental design seems very solid with the very new and interesting models being analyzed.
    - However, I would be interested in how truly autoregressive tabular models are affected by the splits and by your embeddings [1].
    - How are non DL models effected by the splits? I.e. boosting models, or PFNs?
- More importantly, no code is provided during submission. While all results and used benchmarks are very consistent, reproducible code should be provided already during submission time.

---
[1] Thielmann, Anton Frederik, et al. "Mambular: A sequential model for tabular deep learning." arXiv preprint arXiv:2408.06291 (2024).

**Methods And Evaluation Criteria:**

- Yes the benchmarks are very solid, however there are some questions I have regarging the results:
    - Figure 8 on the left you have MLP and MLP-PLR but then have PLR as an embedding strategy. What do you use in MLP-PLR for the other embedding strategies and how does MLP differ from MLP-PLR when using PLR embedding strategy?
    - Since in Figure 8 you analyze the random splits, how does the random split compare in Figure 3?

**Other Comments Or Suggestions:**

typos:
Abstract, line 028 "analyses" should be "analyze"

introduction:
Not just continuous/categorical for regression classification, i.e.distributional approaches, count data/poisson

**Other Strengths And Weaknesses:**

- The paper is overall very well written and the benchmarks/tests seem extremely solid. Models as new as TabM (ICLR 2025) are already included.

**Questions For Authors:**

-  Where are the splits a), b), c) and d) coming from? Is that from Rubachev (2025) or your contribution? and why is "Ours" not shown in the graphic on the right?

**Relation To Broader Scientific Literature:**

The paper very directly relates to the TabRed work proposed by [1] as also noted by the authors. Additionally, the temporal embeddings are also minor adjustments to the work proposed by [2]. The -in my opinion- most interesting finding, that random splits seem to work very well is not adequately analyzed and addressed as it directly confronts the ideas presented in [1].

---
[1] Rubachev, Ivan, et al. "TabReD: Analyzing Pitfalls and Filling the Gaps in Tabular Deep Learning Benchmarks." arXiv preprint arXiv:2406.19380 (2024).
[2] Gorishniy, Yury, Ivan Rubachev, and Artem Babenko. "On embeddings for numerical features in tabular deep learning." Advances in Neural Information Processing Systems 35 (2022): 24991-25004.

**Theoretical Claims:**

- Yes, however there are no proofs/theoretical contributions

---

> ### Author Rebuttal · Authors · 2025-04-01
>
> Thank you for your valuable feedback! We will address your concerns in the following responses.
>
> First, we would like to clarify several key differences between our work and TabReD and PLR.
>
> 1. **Difference from TabReD**: TabReD shows that real-world tabular datasets, inherently containing temporal shifts, require **temporal splits for realistic test sets**. Random splits may lead to misleading assessments, and TabReD demonstrates that temporal splits significantly alter model rankings. Our work builds upon TabReD by identifying **validation set splitting** as a crucial factor affecting model performance once the test set position is fixed. We find that, in this setup, randomly splitting the validation set yields better results than using the temporal split in TabReD. We further analyze the underlying reasons behind this observation and propose our refined temporal split. Therefore, our contribution focuses on **validation set splitting**, whereas TabReD primarily addresses **test set splitting**, making the two approaches complementary rather than overlapping.
> 2. **Difference from PLR emb**: Our temporal emb differs from PLR emb both in **scope and design**. PLR emb is a numerical feature emb method that samples periodicities from $N(0, \sigma)$ to capture cycles in numerical features. In contrast, our temporal emb is specifically designed to incorporate **timestamp information** into models, then treat it as a numerical input feature. It is a **plug-and-play** approach. Structurally, our method relies on **prior cycles**, making it more suitable for handling temporal patterns. Additionally, we explicitly address the challenges of **multi-period coupling** and **trend representation**, which are essential in temporal settings but are not considered in PLR emb. We also **consider PLR emb a baseline** when designing our temporal emb (figure 8 left).
>
> We hope this clarifies our contributions. Our code is available at https://anonymous.4open.science/r/Tabular-Temporal-Shift-BCCA/, with additional results.
>
> > Random split perform identical.
>
> **The random split serves as a baseline**. By analyzing its differences with temporal splits in TabReD, we identified the impact of training lag and validation bias, **motivating our temporal split strategy**.
>
> **While the random split performs well, it suffers from instability** (Std increased by 154%). Our proposed temporal split not only maintains competitive performance but also significantly improves stability, as shown in **Table B in repository**.
>
> > How are non DL/autoregressive/ICL methods effected by the splits?
>
> 1. We have presented the impact of non-DL methods (including Linear, Boosting methods, and Random Forest) under different splits in **Figure 2, Figure 5, and the Appendix (page 13)**. These methods consistently show improvements, with generally better performance in our new temporal split.
> 2. We tested the **Mambular** method. Due to the large size of the TabReD dataset and the inefficiency of autoregressive methods, we only provided results for six datasets. Under our split, its performance showed a significant improvement (+4.56%), as shown in Table E in repository.
> 3. We tested **TabPFN v2**. Since the general model does not require training, we adjusted its context samples: in the Original split, we randomly selected 10,000 context samples, while in Ours, **we selected the 10,000 context samples closest to the test set**. This also led to an improvement (+0.71%), as shown in Table E.
>
> > Since Fig 8 you analyze random splits, how does random split compare in Fig 3?
>
> Figure 8 focuses on comparing **our updated temporal split** with the improvement brought by adding temporal embs. **The random split is only presented in Figures 2 and 5**.
>
> In Figure 3, the bar chart on the right compares the performance of four specifically constructed splits to analyze the effects of reducing training lag, mitigating validation bias, and ensuring validation set equivalence. **Since these splits contain different amounts of data, their results cannot be directly compared to those of the Original, Ours, or Random splits**.
>
> > Where are the splits a), b), c) and d) coming from?
>
> The splits a), b), c), and d) are **entirely our contribution**. Our work focuses on different aspects compared to TabReD.
>
> > Why is "Ours" not shown in the graphic on the right?
>
> The (a,b,c,d) splits are used **only for analysis** of training lag, validation bias, and validation equivalence, which can be seen as an **ablation study**. These splits were created by **discarding data** due to dataset limitations. In contrast, both the original and our temporal splits use the entire dataset before $T_{train}$. As a result, **they cannot be directly compared with the Original or Ours**.
>
> We hope this response addresses your concern. Please feel free to raise any further questions!

---

> > ### Comment · Reviewer_nBxA · 2025-04-01
> >
> > Dear authors,
> >
> > thank you for your answers and clarifications.
> >
> > > How are non DL/autoregressive/ICL methods effected by the splits?
> >
> > Thank you for these experiments. Improvements for TabPFN and even for autoregressive models (although it seems to perform very poorly in general) are very interesting. I would appreciate it, if you included all of these in the paper/appendix.
> >
> > ---
> > As a result of your clarifications/efforts during the rebuttal, I have adjusted my score. 2-> 3

---

> > > ### Author Response · Authors · 2025-04-07
> > >
> > > Thank you for your engagement and effort during the review process! We are also glad that our efforts to address your concerns were helpful. All discussed revisions will be carefully incorporated into the final version.

---

### Official Review · Reviewer_2MA7 · 2025-03-13

**Overall Recommendation:** 4

**Summary:**

The paper investigates the impact of temporal shift in tabular data and presents a set of solutions to mitigate its effects. Since tabular data instances are typically collected in chronological order, temporal shift naturally arises. The authors first find that the commonly used time-based validation split results in worse performance compared to random splitting, and propose a refined temporal splitting protocol designed to reduce training lag and verification bias. Then, from the perspective of model representation, the authors find that the existing methods fail to capture temporal information, and propose a temporal embedding method for deep methods. The experimental results show that both the splitting protocol and the temporal embedding significantly improve the performance of the model under temporal shift scenarios.
## update after rebuttal: Most of my concerns are addressed

**Claims And Evidence:**

Yes, the claims are generally well-supported by experimental results.

**Essential References Not Discussed:**

This paper sufficiently reviews the relevant literature.

**Experimental Designs Or Analyses:**

The experimental design is well-structured and effectively evaluates the proposed claims and methods.

Splitting protocol: Four new splits are used to verify the effects of reducing training lag, reducing validation bias, and ensuring the equivalence of the validation set, respectively, with controlled variables. The use of MMD visualization and loss distribution further supports these claims.

Temporal embedding: The authors first identify the missing temporal information in MLP representations, including multiple periods and trend information. They then propose a temporal embedding that introduces period and trend information into the model. The only potential issue is that the authors do not present the model representation after incorporating their temporal embedding.

**Methods And Evaluation Criteria:**

Yes, the proposed splitting protocol and temporal embedding are all tested on the TabReD benchmark, which focuses on temporal shifts in tabular data.

**Other Comments Or Suggestions:**

There are some typos in the paper. The authors only show the performance improvement of the model after applying the splitting protocol and temporal embedding. It is recommended to include a performance comparison of the methods under different protocols.

**Other Strengths And Weaknesses:**

Strengths:
The paper is well structured and easy to follow.
The experiment designs and results are convincing.

Weaknesses:
The paper compares various types of models (e.g., retrieval-based methods, ensemble-based methods) and provides some analysis in the experimental section, such as “the importance of no-lag candidates for retrieval-based methods.” However, the detailed design for these different methods is not discussed in the related work or preliminary sections.

The authors claim that their splitting protocol achieves similar performance to random splitting while improving stability. However, they present the percentage change in the robustness score, which may not be intuitive.

The performance of the MLP on the HD dataset is significantly weaker than that of the other methods, and the authors excluded this dataset from the percentage improvement calculation for the other methods on the MLP. This could potentially cause confusion. It is recommended to use a metric that is robust to outliers.

**Questions For Authors:**

Please refer to the weaknesses and comments.

**Relation To Broader Scientific Literature:**

Tabular learning is generally based on the i.i.d. assumption, but this paper focuses on temporal shift, which has strong practical significance. Among recent deep tabular models, retrieval-based methods like ModernNCA have demonstrated excellent performance but are considered to perform poorly in distribution shift scenarios. This paper shows that by applying a refined splitting protocol and temporal encoding, retrieval-based methods can regain competitiveness.

**Theoretical Claims:**

This paper focuses on experiment design and analysis and does not make any theoretical claims.

---

> ### Author Rebuttal · Authors · 2025-04-01
>
> We are grateful for your constructive suggestions! We will address your concerns in the following responses.
>
> > The detailed design for these different methods is not discussed.
>
> We apologize for this oversight. We will add an additional section in the preliminary in the revision to introduce the fundamentals of learning from tabular data.
>
> > They present the percentage change in the robustness score, which may not be intuitive.
>
> Thank you for your suggestion! We have now additionally provided the **change in standard deviation** in **Table B** in the repository: https://anonymous.4open.science/r/Tabular-Temporal-Shift-BCCA/. In our comparison, while our method results in a slightly higher standard deviation than the original split (+16.7%), it achieves a significantly lower standard deviation compared to random splitting (+154%).
>
> > It is recommended to use a metric that is robust to outliers.
>
> We agree. In the revision, we will replace the average percentage change calculation with a **robust average**, which excludes the maximum and minimum values when computing the percentage change across the eight datasets. We have updated Figure 2 by **Figure A** in repository, and **it continues to support the same conclusion**: random splitting significantly improves model performance compared to the original split and aligns more closely with existing benchmark results.
>
> > Include a performance comparison of the methods under different protocols.
>
> We have included a **comparison of model performance under different protocols**, specifically after applying our splitting method and temporal embedding, shown in **Figure A bottom**. All comparisons use the **robust average** to compute the average percentage change relative to the original split for MLP, ensuring direct comparison. Additionally, we provide the **average rank** of the models for a more comprehensive evaluation, shown in **Table A**.
>
> We hope this response addresses your concern. Please feel free to raise any further questions!

---

> > ### Comment · Reviewer_2MA7 · 2025-04-03
> >
> > Most of my concerns are solved and I'd like to raise the score.

---

> > > ### Author Response · Authors · 2025-04-07
> > >
> > > Thank you for your insightful comments and suggestions, which have greatly contributed to improving our work! We will incorporate the corresponding changes in the revision.

---

### Official Review · Reviewer_dxDF · 2025-03-14

**Overall Recommendation:** 3

**Summary:**

The paper tackles the problem of how deep tabular methods deteriorate under temporal distribution shifts, where data distributions evolve over time. It demonstrates that typical temporal splitting (training on earlier data, validating on data just slightly more recent, and then testing on even later data) can hinder performance because of a training lag (lack of recent training examples) and validation bias (the validation split may not fully reflect the larger distribution shift faced at test time). By carefully analyzing these issues, the paper proposes a new temporal splitting protocol that reduces training lag and validation bias and thereby achieves performance closer to that of a random data split, while maintaining temporal realism.


## update after rebuttal
After carefully considering the points you presented and other reviewers' comments, I still believe that my initial evaluation remains accurate. Therefore, my score remains unchanged.

**Claims And Evidence:**

Yes. The claims made in the paper are mostly empirically validated with comprehensive experimental results.

**Essential References Not Discussed:**

To the best of my knowledge, the paper should have included the essential references.

**Experimental Designs Or Analyses:**

The experiments appear thorough and carefully controlled, with clear metrics and repeated random seeds. The design supports the main conclusions well.

1. **[Important] Experimental Design:** The authors carefully **ablate different splitting strategies** in terms of training/validation intervals, time-lags, reversed splits, etc. This isolates the roles of lag, bias, and “equivalence” in validating the test distribution.
2. **Qualitative Analysis:** They use **MMD heatmaps** to illustrate how the model’s learned feature distributions differ across time, giving a qualitative sense of whether or not periodic/temporal patterns are preserved.

**Methods And Evaluation Criteria:**

Yes. The temporal shift problem is common and the proposed method seem to resolve it accordingly as expected.

**Other Comments Or Suggestions:**

1. It might help if the authors analyzed the cost of ignoring certain periods or the potential mismatch between a fixed Fourier period (e.g., 7 days) and real data that might have a different cycle.

**Other Strengths And Weaknesses:**

**Strengths**

1. **Insightful diagnosis** of where conventional temporal splits fail: the paper clarifies the subtle difference between a time-based approach that is correct for causal or forecasting tasks vs. real-world tabular tasks that remain partly cross-sectional.
2. **Lightweight method** (Fourier embedding) that is easy to replicate in practice, plus a well-reasoned splitting procedure.
3. **Extensive experimental evidence** with thorough ablations, multi-seed runs, and MMD-based visualizations.

**Weaknesses**

1. **[Important]** The **temporal embedding** approach is tested primarily on a fixed set of known cycles (daily, weekly, monthly, yearly). Real data might have domain-specific cycles or more complex patterns that could require further tuning.
2. The authors do not seem to provide the code, so I remain conservative about the results reported in the paper.

**Questions For Authors:**

1. **[Important]** The authors focus on daily, weekly, monthly, and yearly cycles. Are there practical guidelines for selecting these cycles or discovering new cycles automatically in domain-specific tasks? Have you tried letting the model learn different frequencies if the known cycles are not relevant?
2. **[Important]** As the proposed splitting method do not seem to keep the original temporal order of the data samples, is it possible to cause data leakage problem? For instance, if an older sample is used for validation, and the corresponding newer sample is used in training.

**Relation To Broader Scientific Literature:**

The paper also aligns with broader trends in studying temporal distribution shifts in fields like time-series forecasting and handling “open-environment” learning.

**Theoretical Claims:**

The paper does not seem to make theoretical claims.

---

> ### Author Rebuttal · Authors · 2025-04-01
>
> We appreciate your thoughtful comments! We will address your concerns in the following responses.
>
> > The authors do not seem to provide the code, so I remain conservative about the results reported in the paper.
>
> Our code is now available at https://anonymous.4open.science/r/Tabular-Temporal-Shift-BCCA/. Enjoy the code!
>
> > The temporal emb approach is tested primarily on a fixed set of known cycles.
>
> **The focus of this paper is to analysis why models perform poorly in temporal shift scenarios.** We identified the absence of temporal information in model representations and **improved performance by introducing our temporal embedding, thereby completing our argument**.
>
> Handling unknown or variable cycles will be the focus of **future work**, and we will include this discussion in the **limitations** section. Regarding the approach to learning **unknown or variable cycles**:
>
> - A common approach is to **reweight training samples** [1], but such methods **require an accurate validation set** (instead of a validation set similar to the test time), which is **difficult to obtain in temporal shift tasks**.
> - Another method is to **introduce a matching mechanism** between the training and test set distributions (e.g., attention), but this **requires having a test distribution at test time**, meaning multiple test samples must be obtained, which is closer to test-time adaptation [2].
>
> Based on the MMD visualization and experimental results comparison presented in the paper, we believe the existing fixed cycles already **effectively cover most scenarios (lines 379-384)**. We have also provided experimental results with variable cycles, as addressed in the next question.
>
> > Have you tried letting the model learn different frequencies if the known cycles are not relevant?
>
> Yes, we also experimented with **tuning hyperparameters** to find the cycles, results in **Table C in the repository**. When using fixed prior periods, ModernNCA achieved a 0.30% performance improvement, while **setting adjustable cycles resulted in a -2.48% performance decline**.
>
> In temporal distribution shift scenarios, due to the absence of an entirely accurate validation set, we believe that **prior knowledge of fixed cycles is more stable and interpretable than adjustable cycles**.
>
> It's also important to note that in many tasks, **complete cycles are not available**. For example, in the weather dataset, there is a yearly cycle, but the training set does not span a full year, which highlights the importance of prior knowledge.
>
> > Are there practical guidelines for selecting these cycles or discovering new cycles automatically in domain-specific tasks?
>
> For domain-specific tasks, we still recommend **using prior cycles informed by expert knowledge** or setting them **based on MMD visualization**.
>
> > Is it possible to cause data leakage problems?
>
> This splitting method does not introduce data leakage, as no information that would be unavailable at deployment is used during training. Therefore, **the model's performance on the test set remains reliable**.
>
> We hope this response addresses your concern. Please feel free to raise any further questions!
>
> ---
>
> [1] Mengye Ren et al. Learning to Reweight Examples for Robust Deep Learning. ICML 2018: 4331-4340
>
> [2] Changhun Kim et al. AdapTable: Test-Time Adaptation for Tabular Data via Shift-Aware Uncertainty Calibrator and Label Distribution Handler. CoRR abs/2407.10784

---

### Official Review · Reviewer_k7kB · 2025-03-15

**Overall Recommendation:** 3

**Summary:**

The paper studies temporal shifts aspects of tabular data

The paper has two main contributions:
- First, authors propose a data splitting and validation protocol for improved model performance. The protocol aims to mitigate two phenomena discussed in the paper *1) training lag* (distances between last training timestamp and the first testing timestamp – intuitively the more up-to-date the training data is the better) *2) validation bias* (the difference between `train ↔ val` and `train ↔ test` differences – this can affect model selection in tuing and early stopping)
- Second, the authors look at the temporal patterns in the data (periodicity, trends) and argue that some architectures fail to capture those. To remedy this authors propose a timestamp embedding, which improves performance for some DL architectures.

**Claims And Evidence:**

Claims are adequately (albeit not fully) supported by the experimental evidence.

Here are the key aspects that should be expanded upon in the experiements in my view:

**1**) The first contribution (regarding splitting strategies) makes a claim that the proposed non-standard validation procedure is better for each model, mainly relying on the fact that hold-out future test scores are better.

The authors should be carefull with proposing a new general evaluation procedures (moreover non-standard ones with "backward in time" validation) because 1) better test scores do not necessarily indicate that benchmark (including the protocol) is a better reflection of real-world conditions and model performance (for example, the near-zero train/test gap might be impossible in real-world deployments).

Best solution I see for this issue is – instead of proposing a new better protocol, position the work as an analysis of e.g. train/test gap, or validation bias and expanding such analyses.

**2**) In the second contribution introducing the mechanisms and the utility of the periodic time embedding is not fully studied (e.g. its effects are not universal across model types)

**Essential References Not Discussed:**

No essential references were missed, *but* the above discussion of relation to broader scientific literature points to relevant areas that were not discussed (data splitting strategies with non i.i.d. data in neighboring domains).

**Experimental Designs Or Analyses:**

There are issues in the experimental analyses that need further investigation:

**1**) The result on the Homecredit Default dataset excluded from all comparisons, the exlanation of the sub-par performance provided by authors in the appendix does not seem correct:
> One key difference in the implementation is that the (HD) HomeCredit-Default dataset suffers from severe class imbalance, which makes it difficult for methods with limited feature extraction capabilities, such as MLP, SNN (Klambauer et al., 2017), and DCNv2 (Wang et al., 2021b), to perform well. TabReD utilizes numerical feature encoding, which may significantly improve the performance of these models on this dataset, but the improvement is not substantial on other datasets.

Publicly available TabReD benchmark results, suggest that MLPs without numerical feature embeddings (I interpreted encoding here as – embeddings for numerical features, please correct me if I'm wrong).

Additionally, results on the SH (Sberbank Housing) dataset for the MLP are significantly worth here than the ones reported in the TabReD paper.

Otherwise, the protocol is solid, I don't see other problems.

**Methods And Evaluation Criteria:**

- Comparisons are done model-vise – e.g. all model types individually did improve their test scores, how does new strategies affect our knowledge regarding model comparisons? Does this make results more consistent with prior benchmarks? Plots with relative improvement over the baseline MLP (trained in the same setting as all other methods could be better suited for answering such questions).
- The evaluation done in figure 3 do not consider the confound effects of varying the training set. Some additional experiments where only the validation and test sets are changed to model train/test gaps and validation bias effects would greatly imporve the robustness of results.
- In figure 3. The "ours" variant is not present for the (a,b,c,d) splits of equivalent size.
- Figure 6 analysis is done just for the smiplest MLP model. Results from figure 8 suggest that proposed embedding only helps when using simplest (not SoTA models). This effect is not investigated. Why is it so? Are SoTA models able to learn the temporal patterns without any additional embeddings?

**Other Comments Or Suggestions:**

Mostly about the presentation and writing:
- When describing the experiments with training lag and validation bias, I think a clearer way (this was how I understood this better) is not to communicate the results would be e.g. split (a) has less bias compared to split (c) (not that it is without bias).

**Other Strengths And Weaknesses:**

Strengths:

-  A more in-depth look at handling of temporal non-i.i.d tabular data is timely and interesting, the paper is doing important work (I do not want this review and comments to discourage the ovearll direction the work is taking)
- The analysis and insights into both splitting strategies and other temporal patterns in datasets are new and interesting.
- The writing is good, but some points could be iterated upon and improved (I had trouble digesting the arguments)

**Questions For Authors:**

- Did I understand correctly that in Figure 2. You split on train and val randomly but leave the test out temporally? If so, this should be better communicated.
- What is "one-dimension PLR embedding"?
- Are SoTA models able to learn the temporal patterns without any additional embeddings?

**Relation To Broader Scientific Literature:**

The paper extends observation made in the original TabReD benchmark paper and investigates the aspects of non i.i.d validation.

I find the discussion of similar procedures lacking (that are used in financial data analysis, priror kaggle competitons).

Examples of relevant work in this area:
- "A survey of cross-validation procedures for model selection" https://arxiv.org/abs/0907.4728 (Section 8.3)
- "CVTT: Cross-Validation Through Time" https://arxiv.org/abs/2205.05393 (and related work mentioned there)
- Purged K-Fold cross validation, described in from "Advances in financial Machine Learning (2018)" is often used in competitons when "backtesting". It includes validation at different time periods.

**Theoretical Claims:**

There are no theoretical claims.

---

> ### Author Rebuttal · Authors · 2025-04-01
>
> Thank you for your insightful feedback! We will address your concerns in the following responses.
>
> > The authors should be carefull ...
>
> Our experimental setup **strictly follows TabReD**. Moreover, since this is a temporal scenario where each test sample is evaluated individually, the **varying time gaps between each sample and training set** reflect real-world conditions.
>
> Additionally, **this does not impact our training protocol**, which only provides **guidance stated in lines 307–316**. Extensive experiments have validated the robustness of our training protocol, even when the training lag cannot be reduced to zero (**(c) vs. (d)** in Figure 3).
>
> The newly proposed temporal split is **primarily intended to illustrate the effectiveness of this protocol**, reinforcing our focus on **analyzing** training lag and validation bias.
>
> > Temporal emb is not fully studied. Hardly helps SOTA.
>
> This issue is already identified in our paper (**lines 401, 410**). Our temporal emb **converts timestamps into numerical inputs**, which may be **incompatible with PLR emb**. Specifically, once timestamps are embedded, their representation reflects temporal similarity. Applying another periodic transformation via PLR could **increase optimization difficulty**.
>
> Directly feed the temporal emb into the model backbone **consistently improves**: MLP-PLR +0.01% → +0.26%, TabM +0.07% → +0.15%, MNCA +0.30% → +0.38%. Please refer to Table D in the repository: https://anonymous.4open.science/r/Tabular-Temporal-Shift-BCCA/.
>
> > How does new strategies affect model comparisons?
>
> Please refer to our response to Reviewer 2MA7.
>
> > Fig 3 does not consider the confound effects ...
>
> Our current experiments **already addressed this issue**. Our analysis of the loss distribution (**Fig 4 right**) isolates the impact of validation and test set variations while fixing the training set.
>
> In splits (a,c), with the training set fixed earlier in time and shifts in validation and test sets, we observe in **lines 234-251** the impact of reducing training lag and validation bias.
>
> >The Ours variant is not present for the (a,b,c,d) splits of equivalent size in fig 3.
>
> The (a,b,c,d) splits are used **only for analysis** of training lag, validation bias, and equivalence. These splits were created by **discarding data** due to dataset limitations. In contrast, both the original and our temporal splits use the entire dataset before $T_{train}$.
>
> > Figure 6 done just for MLP.
>
> We will add visualizations for SOTA model representations.
>
> > The result on HD excluded from all comparisons.
>
> **HD dataset was excluded only from the comparison in Fig 2**. This is because Figure 2 calculates the improvement of other methods relative to MLP, and MLP performs poorly on the HD dataset. As a result, the relative improvement is significantly larger (~80%) compared to the average improvement on other datasets (<6%). Including this dataset in the mean obscures the result.
>
> However, all other comparisons do not rely on relative improvements over MLP, **so the HD dataset is included in all other results**. Suggested by Reviewer 2MA7, we will adopt **robust average** to compute the mean improvement in the revision to better handle such cases.
>
> > TabReD suggest that MLPs without numerical embs ...
>
> Here we are not referring to **numerical embs during training**, but **feature encoding during preprocessing**, such as noisy-quantile encoding. This will be clarified in the revision.
>
> This aspect is not mentioned in the TabReD paper but can be inferred from the code:
>  https://github.com/yandex-research/tabred/blob/main/exp/mlp/homecredit-default/tuning.toml#L54, which assigns noisy-quantile and ordinal for the MLP method on the HD dataset. **TabReD assigns different encodings to different method-dataset pairs**.
>
> We believe this introduces fairness inconsistencies. Special encodings (e.g., noisy-quantile) do not always improve performance. Therefore, we used only basic encoding (**none for numerical and one-hot for categorical features**) for a fair comparison, despite lowering MLP’s performance on the HD dataset. This explains the difference between our results and those in the TabReD paper.
>
> > Did I understand Fig 2 correctly ?
>
> Yes. We will further clarify this.
>
> > What is 1D PLR?
>
> Sorry for this confusion. Our proposed temporal emb is specifically designed for the timestamp, we choose a **PLR emb applied to the single timestamp input** as a baseline.
>
> > Are SoTA models able to learn ...
>
> Without timestamp information, models cannot learn the order and periodicity of samples. **The challenge is incorporating temporal information**. Fig 8 shows that treating timestamps as numerical features causes performance drop, aligning with the common practice of removing timestamps. Only by combining temporal emb with periodicity knowledge can temporal patterns be effectively incorporated.
>
> Due to the limit, some responses may lack detail. Please feel free to raise any further concerns!

---

> > ### Comment · Reviewer_k7kB · 2025-04-05
> >
> > Thanks for the extensive rebuttal response.
> >
> > I encourage authors to clarify the aspects discussed during the review period in the revision (especially key differences in protocol like not using numerical feature normalization).
> >
> > I also encourage to more clearly report results for the temporal embeddings in the SoTA architectures. In particular the aspect that it does not play well with numerical feature embedings but that this can be fixed (plus the addition of state-of-the-art models in figure 6).
> >
> > I remain a bit sceptical regarding the universality of the proposed splitting procedure (as a go-to recommendation for future research) but still find the result interesting for the community.
> >
> > Many of my initial concerns have been addressed by the rebuttal. I raised the score accordingly.

---

> > > ### Author Response · Authors · 2025-04-07
> > >
> > > We deeply appreciate your constructive suggestions and valuable feedback throughout the review process, which have greatly helped us improve our work! The changes addressed during the rebuttal will be carefully reflected in our revised version. Thank you again for your time and encouragement!

---

### Decision · Program_Chairs · 2025-05-01

**Decision:**

Accept (poster)

**Comment:**

This paper explores the impact of temporal shifts in tabular data and identifies two critical factors—training lag and validation bias—that hinder generalization. To address these, the authors propose a refined data-splitting protocol and introduce a lightweight temporal embedding module that enhances temporal representation learning. These two contributions lead to consistent performance improvements across a variety of tabular methods on the TabReD benchmark.

The paper is well-written and addresses a practically important yet under-explored challenge. Its strengths include novel insights into common evaluation pitfalls, broad empirical validation across diverse models and datasets, and a temporal embedding that is simple, effective, and easily integrable with existing tabular architectures—especially benefiting non-temporal models.

Some reviewers raised concerns regarding the generalizability of the proposed splitting strategy, the relatively smaller gains of the temporal embedding on more complex models, and its reliance on prior knowledge of temporal patterns. The authors provided a comprehensive rebuttal, effectively addressing most of these concerns. While the proposed splitting strategy may not be universally applicable, it draws valuable attention to the importance of evaluation protocol design in temporally shifting environments and offers a practical baseline for future work. Following the discussion, reviewer consensus improved, resulting in final scores of 3, 3, 3, and 4.

Overall, the paper makes a meaningful and empirically grounded contribution to deep tabular learning under temporal shift. I recommend acceptance.